Widespread air pollutants of the North China Plain during the Asian summer monsoon season:
A case study
Jiarui Wu[1,3], Naifang Bei[2], Xia Li[1,3], Junji Cao[1*], Tian Feng[1], Yichen Wang[1], Xuexi Tie[1], and Guohui Li[1*]
[1]Key Lab of Aerosol Chemistry and Physics, SKLLQG, Institute of Earth Environment, Chinese Academy of
Sciences, Xi'an, China
[2]School of Human Settlements and Civil Engineering, Xi'an Jiaotong University, Xi'an, Shaanxi, China
[3]University of Chinese Academy of Science, Beijing, China
[*]Correspondence to: Guohui Li (ligh@ieecas.cn) and Junji Cao (jjcao@ieecas.cn)
**Abstract**: During the Asian summer monsoon season, prevailing southeasterly -
southwesterly winds are subject to delivering air pollutants from the North China Plain (NCP)
to the Northeast and Northwest China. In the present study, the WRF-CHEM model is used to
evaluate contributions of trans-boundary transport of the NCP emissions to the air quality in
the Northeast and Northwest China during a persistent air pollution episode from 22 to 28
May 2015. The WRF-CHEM model generally performs well in capturing the observed
temporal variation and spatial distribution of fine particulate matters ($PM_{2.5}$), ozone ($O_3$), and
$NO_2$. The simulated temporal variation of aerosol species is also in good agreement with
measurements in Beijing during the episode. Model simulations show that the NCP emissions
contribute substantially to the $PM_{2.5}$ level in Liaoning and Shanxi provinces, the adjacent
downwind areas of the NCP, with an average of 24.2 and 13.9 $\mu g\ m^{-3}$ during the episode,
respectively. The $PM_{2.5}$ contributions in Jilin and Shaanxi provinces are also appreciable,
with an average of 9.6 and 6.5 $\mu g\ m^{-3}$, respectively. The average percentage contributions of
the NCP emissions to the $PM_{2.5}$ level in Liaoning, Jilin, Shanxi, Shaanxi provinces are 40.6%,
27.5%, 32.2%, and 20.9%, respectively. The NCP emissions contribute remarkably to the $O_3$
level in Liaoning province, with an average of 46.5 $\mu g\ m^{-3}$, varying from 23.9 to 69.5 $\mu g\ m^{-3}$.
The $O_3$ level in Shanxi province is also influenced considerably by the NCP emissions, with
an average contribution of 35.1 $\mu g\ m^{-3}$. The $O_3$ level in Shanxi province is also influenced
considerably by the NCP emissions, with an average contribution of 35.1 $\mu g\ m^{-3}$. The average
$O_3$ contributions of the NCP emissions to Jilin and Shaanxi provinces are 28.7 and 20.7 $\mu g$
$m^{-3}$, respectively. The average percentage contributions of the NCP emissions to the
afternoon $O_3$ level in Liaoning, Jilin, Shanxi, and Shaanxi provinces are 27.4%, 19.5%,
21.2%, and 15.8%, respectively. However, the effect of the NCP emissions on the air quality
in Inner Mongolia is generally insignificant. Therefore, effective mitigations of the NCP
emissions not only improve the local air quality, but also are beneficial to the air quality in
the Northeast and Northwest China during the Asian summer monsoon season.

## 1 Introduction

With the rapid growth of industrialization, urbanization and transportation, China has experienced severe air pollution with high levels of fine particulate matters ($PM_{2.5}$) and ozone ($O_3$) recently (e.g., Chan and Yao, 2008; Zhang et al., 2013; Kurokawa et al., 2013; Li et al., 2017b). Although the Chinese State Council has issued the 'Atmospheric Pollution Prevention and Control Action Plan' in September 2013 with the aim of improving China's air quality, heavy haze or photochemical smog still frequently plagues China, especially in the North China Plain (NCP),Yangtze River Delta (YRD), and Pearl River Delta (PRD). Elevated $O_3$ and $PM_{2.5}$ concentrations in the atmosphere not only perturb regional and global climates, also exert adverse effects on air quality, ecosystems, and human health (Weinhold, 2008; Parrish and Zhu, 2009).

The NCP has become one of the most polluted areas in the world due to a large amount of emissions of pollutants and frequent occurrence of unfavorable meteorological situations, as well as the topography (e.g., Tang et al., 2012; Zhang et al., 2013; Zhuang et al., 2014; Pu et al., 2015; Long et al., 2016). Heavy haze with extremely high $PM_{2.5}$ concentrations often covers the NCP during wintertime, partially attributable to the coal combustion for domestic heating (e.g., He et al., 2001; Cao et al., 2007; H. Li et al., 2017a). However, even in summer, with improvement of the evacuation condition and increase of precipitation, photochemical smog with high levels of $PM_{2.5}$ and $O_3$ still engulfs the NCP (e.g., Gao et al., 2011; Hu et al., 2014; Cao et al., 2015; Wu et al., 2017). The $PM_{2.5}$ concentrations during summertime in the NCP are generally lower than those in winter, but still much higher than 35 μg m$^{-3}$, the first grade of National Ambient Air Quality Standards (NAAQS) in China (Feng et al., 2016; Z. S. Wang et al., 2016; Sun et al., 2016). The average summertime $PM_{2.5}$ concentrations in the NCP are 77.0 ± 41.9 μg m$^{-3}$ in 2013, much more than those in other regions of China and also exceeding the second grade of NAAQS (Hu et al., 2014). In addition, increasing $O_3$

precursors emissions has caused serious $O_3$ pollution during summertime in the NCP (e.g.,

Zhang et al., 2009; Xu et al., 2011; Kurokawa et al., 2013). Li et al. (2017b) have reported

that the maximum 1h $O_3$ concentration exceeds 200 μg m$^{-3}$ in almost all the cities in Eastern

China from April to September 2015, mainly concentrated in the NCP and YRD, showing a

widespread and persistent $O_3$ pollution. Ma et al. (2016) have found a growth trend of the

surface $O_3$ at a rural site in the NCP from 2003 to 2015, with an average rate of 1.13 ppb per

year. Wu et al. (2017) have shown that the average afternoon $O_3$ concentration in the summer

of 2015 in Beijing is about 163 μg m$^{-3}$.

China is located in a large monsoon domain, and the Asia summer monsoon (ASM)

tends to substantially influence the distribution and trans-boundary transport of air pollutants

in China. Zhu et al. (2004) have proposed that the summertime high $O_3$ concentration over

Western China is due to the monsoonal transport from Eastern China and long-range

transport from South/central Asia and even Europe. Zhao et al. (2010) have also indicated

that $O_3$ transported from South/Central Asia to Western China increases from May to August

because of the northward movement of the India summer monsoon. Huang et al. (2015) have

suggested that an earlier onset of the ASM would lead to more $O_3$ enhancement in the lower

troposphere over the NCP in later spring and early summer. Numerous studies have also

reported that the strength and tempo-spatial extension of the ASM influences the spatial and

temporal distribution of aerosol mass concentrations over Eastern China (Cao et al., 2015; Li

et al., 2016; Cheng et al., 2016). For example, Zhang et al. (2010) have emphasized that the

East ASM plays a major role in determining the seasonal and interannual variations of the

$PM_{2.5}$ concentration over Eastern China. Using the GEOS-CHEM model, Zhu et al. (2012)

have shown that the weakening of the ASM increases the aerosol concentration in Eastern

China. Wu et al. (2016) have pointed out that the regional transport and tempo-spatial

distribution of air pollutants are directly influenced by the East Asian monsoon at seasonal,

**90** inter-annual, and decadal scales.

**91** During the ASM season, meteorological conditions are characterized by prevailing

**92** southwesterly-southeasterly winds over Eastern China. Air pollutants originated from the

**93** NCP are likely to be transported northwards and affect the air quality in its downwind areas,

**94** so it is imperative to quantitatively evaluate the effect of the NCP emissions on the air quality

**95** in its neighboring regions. Previous studies have concentrated on the composition,

**96** characteristics, and sources of the air pollutants over the NCP (e.g., Han et al., 2006; Liu et

**97** al., 2012; Zhao et al., 2013; Li et al., 2015). However, few studies have been performed to

**98** investigate the effect of trans-boundary transport of air pollutants originated from the NCP on

**99** the air quality in the Northeast and Northwest China under the prevailing southerly wind

**100** associated with the ASM.

**101** In this study, we first analyze the role of synoptic situations during the ASM (from May

**102** to September) in the trans-boundary transport over Northern China and further evaluate the

**103** contribution of trans-boundary transport of pollutants originated from the NCP to the air

**104** quality in the Northeast and Northwest China using the WRF-CHEM model. The model

**105** configuration and methodology are described in Section 2. Analysis results and discussions

**106** are presented in Section 3, and conclusions are given in Section 4.

**107**

**108** **2  Model and Methodology**

**109** **2.1 WRF-CHEM Model and Configuration**

**110** A persistent air pollution episode with high levels of $PM_{2.5}$ and $O_3$ from 22 to 28 May

**111** 2015 in Northern China is simulated using the WRF-CHEM model which is developed by Li

**112** et al. (2010, 2011a, b, 2012) at the Molina Center for Energy and the Environment. Table 1

**113** provides detailed model configurations and Figure 1 shows the WRF-CHEM model

**114** simulation domain. It is worth noting that the horizontal resolution of 10 km adopted in this

 Further description of the model is presented in Supplementary Information (SI).

The key characteristics of the aerosol pollution in China are frequently associated with rather efficient secondary formation, including aerosol nucleation and rapid growth under favorable conditions (Zhang et al., 2012; Qiu et al., 2013; Guo et al., 2014; Zhang et al., 2015). The new particle production rate in the WRF-CHEM model is calculated due to the binary nucleation of $H_2SO_4$ and $H_2O$ vapor. The nucleation rate is a parameterized function of temperature, relative humidity, and the vapor-phase $H_2SO_4$ concentration, following the work of Kulmala et al. (1998), and the new particles are assumed to be 2.0 nm diameter. Recent studies have shown that organic vapors are involved in the nucleation process (Zhang et al., 2012) and further studies need to be conducted to consider the contributions of organic vapors to the nucleation process. The secondary organic aerosol (SOA) formation is simulated using a non-traditional SOA model including the volatility basis-set modeling method in which primary organic components are assumed to be semi-volatile and photochemically reactive and are distributed in logarithmically spaced volatility bins (Li et al., 2011a). The contributions of glyoxal and methylglyoxal to the SOA formation are also included in the SOA module. The SOA formation from glyoxal and methylglyoxal is parameterized as a first-order irreversible uptake by aerosol particles, with a reactive uptake coefficient of $3.7 \times 10^{-3}$ for glyoxal and methylglyoxal (Zhao et al., 2006). The simulation of inorganic aerosols in the WRF-CHEM model adopts the ISORROPIA Version 1.7 (Nenes et al., 1998).

For the discussion convenience, Northern China is divided into 3 regions (Figure S1): 1) the North China Plain (including Beijing, Tianjin, Hebei, Shandong, Henan, the south of Jiangsu and Anhui, hereafter referred to as the NCP), 2) the Northeast China (including

**140** Heilongjiang, Jilin, Liaoning and the east part of Inner Mongolia, hereafter referred to as the

**141** NEC), and 3) the Northwest China (including Shanxi, Shaanxi and the west part of Inner

**142** Mongolia, hereafter referred to as the NWC). During the episode, the observed average daily

**143** $PM_{2.5}$ concentration was 75.5 μg m$^{-3}$ and the mean $O_3$ concentration in the afternoon was

**144** 151.2 μg m$^{-3}$ in the NCP. Figure S2 presents the distributions of the anthropogenic emission

**145** rates of volatile organic compounds (VOCs), nitrogen oxide ($NO_x$), organic carbon (OC), and

**146** $SO_2$ in Mainland China, showing that the high emission rates of VOCs, $NO_x$, OC, and $SO_2$

**147** are generally concentrated in the NCP. It is worth noting that uncertainties in the emission

**148** inventory used in this study are rather large considering the rapid changes in anthropogenic

**149** emissions that are not fully reflected in the current emission inventory and the complexity of

**150** pollutants precursors.

**151** **2.2 Data and Methodology**

**152** In the present study, the model performance is validated using the hourly measurements

**153** of $O_3$, $NO_2$, and $PM_{2.5}$ concentrations released by the China's Ministry of Environment

**154** Protection (China MEP), which can be accessed at http://www.aqistudy.cn/. In addition, the

**155** simulated submicron sulfate, nitrate, ammonium, and organic aerosols are compared to the

**156** measurements by the Aerodyne Aerosol Chemical Speciation Monitor (ACSM), which was

**157** deployed at the National Center for Nanoscience and Technology (NCNST), Chinese

**158** Academy of Sciences in Beijing (Figure 1). The observed mass spectra of organic aerosols

**159** are analyzed using the Positive Matrix Factorization (PMF) technique and four components

**160** are identified: hydrocarbon-like organic aerosol (HOA),cooking organic aerosol (COA),coal

**161** combustion organic aerosol (CCOA), and oxygenated organic aerosol (OOA). HOA, COA,

**162** and CCOA are interpreted as a surrogate of primary organic aerosols (POA), and OOA is a

**163** surrogate of secondary organic aerosols (SOA). Furthermore, the reanalysis data from the

European Centre for Medium-Range Weather Forecasts (ECMWF) are used to analyze the
synoptic patterns during the ASM season from May to September 2015.
The mean bias (*MB*), root mean square error (*RMSE*) and the index of agreement (*IOA*)
are utilized to evaluate the performance of the WRF-CHEM model simulations against
measurements. To assess the contributions of the NCP emissions to the near-surface
concentrations of $O_3$ and $PM_{2.5}$ in the NEC and NWC, the factor separation approach (FSA)
is used in this study (Stein and Alpert, 1993; Gabusi et al., 2008; Li et al., 2014). The detailed
description of methodology can be found in SI-2.

**3    Results and Discussion**
**3.1    Synoptic Patterns during the ASM Season**
The ASM commences to prevail from May to September each year in China, with strong
winds blowing from oceans to Eastern China and bringing warm and moist airflow to the
continent. Furthermore, the Western Pacific subtropical high gradually intensifies, and moves
from south to north to influence the weather and climate over China, also transporting water
vapor from the sea to Eastern China. During the ASM season, due to the influence of the
Western Pacific subtropical high, rain belts and associated deep convections move from
Southeastern China to Northern China (Ding et al., 1992, 2005; Lau et al., 1988, 1992; Kang
et al., 2002). Figure 2 shows the geopotential heights at 500 hPa and mean sea level pressure
with wind vectors during the ASM season in 2015. At 500 hPa, the main part of subtropical
high, which is represented by the scope of the contour of 5880 geopotential meter, is located
in Northwest Pacific Ocean. The mean ridgeline of the Western Pacific subtropical high is
located at 25°N, moving from south to north from May to September, which substantially
affects the synoptic conditions in China. Flat westerly wind at 500 hPa prevails over the NCP
and its surrounding regions, indicating a stable weather condition. The mean sea level
pressure shows that most of areas in the NCP are continually influenced by the ASM and the
high-pressure system centering in the Western Pacific, causing the prevailing southeasterly -
southwesterly wind over the NCP and its surrounding areas. The detailed description of the
synoptic conditions during the study episode can be found in SI.
In the region controlled by the Western Pacific subtropical high, a subsidence airflow is
dominant with calm or weak winds, and the temperature is extremely high due to the strong
sunlight, which is favorable for the accumulation and formation of air pollutants. The air
pollutants are likely to be transported from south to north under the persistent effect of
southerly winds.
Figures 3 and 4 present the relationship of $PM_{2.5}$ and $O_3$ concentrations in the NCP with
those in the NEC and NWC during the ASM season from 2013 to 2016, respectively. The
observed $PM_{2.5}$ and $O_3$ concentrations in the NCP exhibit a positive correlation with those in
the NEC and NWC, with the correlation coefficients generally exceeding 0.55. There are two
possible reasons for the positive correlation of $PM_{2.5}$ and $O_3$ concentrations between the NCP
and its surrounding regions. One is that when the NCP and its neighboring areas are
controlled by the same large-scale synoptic pattern, the concentrations of air pollutants
generally vary synchronously. The other is the trans-boundary transport of air pollutants
originated from the NCP to its surrounding areas due to the southerly wind associated with
the ASM. The correlation coefficients of $PM_{2.5}$ and $O_3$ concentrations in the provinces of the
NEC with those in the NCP generally decrease from south to north, with the coefficients of
0.69, 0.56 and 0.52 for $PM_{2.5}$, and of 0.86, 0.76, and 0.76 for $O_3$ in Liaoning, Jilin and
Heilongjiang, respectively. The decreasing trend of the correlation coefficients also exists
from east to west in the NWC, with coefficients of 0.69 and 0.62 for $PM_{2.5}$, and 0.87 and 0.84
for $O_3$ in Shanxi and Shaanxi, respectively. Hence, when severe air pollution occurs in the
NCP in summer, the air quality in its adjacent provinces is likely to be deteriorated, possibly

**214** caused by the trans-boundary transport of air pollutants originated from the NCP.

**215** It is worth noting that the intensity of ASM substantially influences the temporal

**216** variation and spatial distribution of air pollutants (Wu et al., 2016). The East Asia summer

**217** monsoon index proposed by Zhang et al. (2003) is defined as a difference of anomalous zonal

**218** wind between the (10°-20°N, 100°-150°E) and (25°-35°N, 100°-150°E) at 850hPa during

**219** summer (June-August). The year of monsoon index greater than or equal to 2 is defined as

**220** the strong summer monsoon year, and the year of monsoon index less than or equal to -2 is

**221** defined as the weak summer monsoon year. The monsoon index calculated by China

**222** Meteorological Administration shows that the intensity of the summer monsoon in 2015 is

**223** close to the normals (SI-Figure S5).

**224** **3.2 Model performance**

**225** **3.2.1 $PM_{2.5}$, $O_3$ and $NO_2$ Simulations in Northern China**

**226** Figure 5 shows the temporal variations of observed and simulated near-surface $PM_{2.5}$,

**227** $O_3$ and $NO_2$ concentrations averaged over monitoring sites in Northern China. The

**228** WRF-CHEM model generally simulates well the diurnal variation of $PM_{2.5}$ concentrations in

**229** Northern China, with *IOA* of 0.91. The model successfully reproduces the temporal variations

**230** of surface $O_3$ concentrations compared with observations in Northern China, e.g., peak $O_3$

**231** concentrations in the afternoon due to active photochemistry and low $O_3$ concentrations

**232** during nighttime caused by the $NO_x$ titration, with *IOA* of 0.98. However, the model

**233** underestimation still exists in simulating the $O_3$ concentration, with a *MB* of -4.0 μg m$^{-3}$. The

**234** model also reasonably yields the $NO_2$ diurnal profiles, but frequently overestimates the $NO_2$

**235** concentrations in the late evening due to the simulated low PBL height, and underestimates

**236** the concentration in the early morning because of the uncertainties in the $NO_x$ emissions. The

**237** further analysis of the model performance of $PM_{2.5}$, $O_3$ and $NO_2$ concentrations in Northern

**238** China can be found in SI.

### 3.2.2 Aerosol Species Simulations in Beijing

Figure 6 presents the temporal variations of simulated and observed aerosol species at NCNST site in Beijing from 22 to 28 May 2015. Generally, the WRF-CHEM model predicts reasonably the temporal variations of the aerosol species against the measurements, especially for POA and nitrate aerosol, with $IOA$s of 0.81 and 0.90, respectively. The model has difficulties in well simulating the SOA concentrations, with the $IOA$ and $MB$ of 0.69 and -3.6 μg m$^{-3}$, respectively. It is worth noting that many factors influence the SOA simulation, including measurements, meteorology, precursors emissions, SOA formation mechanisms and treatments (Bei et al., 2012, 2013). The model reasonably tracks the temporal variation of the observed sulfate concentration, but the bias is still large, and the $MB$ and $IOA$ are -1.2 μg m$^{-3}$ and 0.68, respectively. The sulfate source in the atmosphere is various, including $SO_2$ gas-phase oxidations by hydroxyl radicals (OH) and stabilized criegee intermediates (sCI), aqueous reactions in cloud or fog droplets, and heterogeneous reactions on aerosol surfaces, as well as direct emissions from power plants and industries (G. H. Li et al., 2017a). G. Wang et al. (2016) have also reported that the aqueous oxidation of $SO_2$ by $NO_2$ is important to the efficient sulfate formation. Considering that the model fails to well resolve convective clouds due to the 10km horizontal resolution, the sulfate formation from the cloud process is generally underestimated. Additionally, large amount of $SO_2$ is emitted from point sources, such as power plants or agglomerated industrial zones, which is much more sensitive to wind fields simulations (Bei et al., 2010). The model performs reasonably well in simulating the ammonium aerosol, with the $IOA$ and $MB$ of 0.77 and -0.4 μg m$^{-3}$, respectively.

### 3.2.3 Simulations of the Spatial Distribution of $PM_{2.5}$ and $O_3$ Concentrations

The peak $PM_{2.5}$ concentration generally occurs from 08:00 to 10:00 Beijing Time (BJT) due to the rush hour. Figure 7 provides the distributions of calculated and observed near-surface $PM_{2.5}$ concentrations along with the simulated wind fields at 08:00 BJT from 23

**264** to 28 May 2015. The calculated $PM_{2.5}$ spatial patterns generally agree well with the

**265** observations at the monitoring sites. The NCP experiences severer $PM_{2.5}$ pollution than its

**266** surrounding areas, with $PM_{2.5}$ concentrations frequently exceeding 150 μg m$^{-3}$ in the

**267** Beijing-Tianjin-Hebei region. During the study episodes, the pollutants are likely to be

**268** transported to the NEC and NWC under the prevailing southwesterly or southeasterly winds

**269** in Northern China, causing the $PM_{2.5}$ concentrations in most of areas of the NEC and NWC

**270** frequently to be higher than 75 μg m$^{-3}$.

**271** The $O_3$ concentration during summertime generally reaches its peak from 14:00 to 16:00

**272** BJT in Northern China (Figure 5). Figure 8 shows the spatial distribution of calculated and

**273** measured near-surface $O_3$ concentrations at 14:00 BJT from 23 to 28 May 2015, along with

**274** the simulated wind fields. Generally, the simulated $O_3$ spatial patterns are consistent with the

**275** observations, but the model overestimation or underestimation still exists. The simulated high

**276** $O_3$ concentrations at 14:00 BJT, exceeding 200 μg m$^{-3}$, are frequently concentrated in the

**277** NCP, which is also consistent with the measurements. The $O_3$ transport to the NEC and NWC

**278** from the NCP is obvious when the winds are southeasterly or southwesterly, inducing the

**279** severe $O_3$ pollution in the NEC and NWC.

**280** In general, the simulated variations of $PM_{2.5}$, $O_3$, $NO_2$ and aerosol species are in good

**281** agreement with observations, indicating that the simulations of meteorological conditions,

**282** chemical processes and the emission inventory used in the WRF-CHEM model are

**283** reasonable, providing a reliable base for the further investigation.

**284** **3.3 Effects of the NCP Emissions on the Air Quality in the NEC and NWC**

**285** To evaluate the contribution of the NCP emissions to the air quality in its neighboring

**286** areas, four model simulations are performed, including $f_{NS}$ with all anthropogenic

**287** emissions from the NCP and non-NCP areas, $f_N$ with anthropogenic emissions from the

**288** NCP only, $f_S$ with anthropogenic emissions from the non-NCP areas only, and $f_0$ without

all anthropogenic emissions. Consequently, the air pollutants concentrations in the NEC and NWC can be separated into four components, including contributions from the local emissions ($f'_S$, $f_S - f_0$), the trans-boundary transport of the NCP emissions ($f'_N$, $f_N - f_0$), the interactions of these two emissions ($f'_{NS}$, $f_{NS} - f_N - f_S + f_0$) and the background ($f_0$).

In the present study, the effect of the NCP emissions on the $PM_{2.5}$ and $O_3$ concentrations in the NEC and NWC is evaluated, considering that they have the long lifetime of several days in the troposphere and often constitute the primary air pollutant during summertime (Seinfeld and Pandis, 2006). However, the trans-boundary transport of $PM_{10}$ is not considered due to its short lifetime of several hours caused by the dry deposition and gravity and the fact that $PM_{10}$ is generally confined to its source region when the wind is not strong enough (Sun et al., 2006).

**3.3.1 Contributions of the NCP Emissions to $PM_{2.5}$ Levels in the NEC and NWC**

Figure 9 shows the simulated spatial distribution of daily mean $PM_{2.5}$ concentrations contributed by the NCP emissions in the NEC and NWC from 23 to 28 May 2015. The contribution of trans-boundary transport of the NCP emissions to the $PM_{2.5}$ concentration is remarkable in Liaoning, frequently exceeding 30 μg m$^{-3}$ in most areas of the province during the episode. The NCP emissions also considerably influence the $PM_{2.5}$ concentration in Jilin, contributing 5~30 μg m$^{-3}$ in most areas and occasionally exceeding 40 μg m$^{-3}$. The effect of the NCP emissions on the $PM_{2.5}$ level in Shanxi and Shaanxi is increasingly evident from 23 to 28 May 2015, with the contribution of up to 50~60 μg m$^{-3}$ in southeast of Shanxi and to a lesser extent of 30~40 μg m$^{-3}$ in the middle part of Shaanxi on 27- 28 May. The contribution of trans-boundary transport of the NCP emissions to the $PM_{2.5}$ level in Inner Mongolia is not significant, which may be attributed to the location of the low pressure and terrain characteristics. Obviously, the effect of trans-boundary transport shows a stepwise

characteristic: the closer to the NCP emission sources, the more remarkable the impact on the
downwind areas. As a result, Liaoning and Shanxi provinces are substantially influenced by
the NCP emissions, while Jilin and Shaanxi provinces are affected to a lesser extent.
The impact of the NCP emissions on the daily average $PM_{2.5}$ concentration in the NEC
and NWC from 22 to 28 May 2015 is summarized in Table 2. On average, the NCP emissions
increase the $PM_{2.5}$ concentrations by 24.2, 9.6, 13.9, 6.5, and 2.6 $\mu g\ m^{-3}$ in Liaoning, Jilin,
Shanxi, Shaanxi, and Inner Mongolia, with the average percentage contribution of 40.6%,
27.5%, 32.2%, 20.9%, and 16.7%, respectively. Figure 10 shows the episode-averaged $PM_{2.5}$
percentage contribution from the NCP emissions to the surrounding areas. The NCP
emissions markedly affect the air quality in Liaoning, accounting for around 20%-50% of the
$PM_{2.5}$ concentration during the episode and with the most substantial impact on the west part
of the province. The NCP emissions contribute about 15%-30% of the $PM_{2.5}$ concentration in
Jilin. Shanxi province is also remarkably affected by the NCP emissions, with more than 25%
of $PM_{2.5}$ concentration contributed by the NCP emissions in most areas. Although Shaanxi
province is a little far from the NCP, the NCP emissions still contribute about 10%-35% of
the $PM_{2.5}$ concentration. The NCP emissions also enhance the $PM_{2.5}$ concentration by 5-50%
in the southern edge of Inner Mongolia, which is adjacent to the NCP.
**3.3.2 Contributions of the NCP Emissions to $O_3$ Concentrations in the NEC and NWC**
Figure 11 shows the simulated spatial distribution of the average afternoon $O_3$
concentrations contributed by the NCP emissions from 23 to 28 May 2015. Similar to the
$PM_{2.5}$ case, the contribution of the NCP emissions to the $O_3$ formation in Liaoning and Jilin
province is increasingly enhanced during the episode (except on 26 May), and on 25 and 27
May, the NCP emissions account for more than 70 $\mu g\ m^{-3}$ of the $O_3$ concentration in most
areas of Liaoning. On 25 and 28 May, the NCP emissions contribute more than 70 $\mu g\ m^{-3}$ of
the $O_3$ concentration in some regions of Jilin. A less impact of the NCP emissions on Jilin

province on 26 May is due to the weakening of the low pressure. The NCP emissions play a progressively important role in $O_3$ concentrations in Shanxi and Shaanxi provinces during the episode, especially on 27 and 28 May when the contribution can be up to 60 μg m$^{-3}$. The impact of the NCP emissions on $O_3$ concentrations in Inner Mongolia is insignificant overall.

Table 3 summarizes the effects of the NCP emissions on the average afternoon $O_3$ concentration in the NEC and NWC from 22 to 28 May 2015. During the episode, the NCP emissions substantially influence the $O_3$ level in Liaoning province, and the afternoon $O_3$ contribution is about 46.5 μg m$^{-3}$ on average, ranging from 23.9 to 69.5 μg m$^{-3}$. The NCP emissions also contribute an average of 28.7 μg m$^{-3}$ to the $O_3$ concentration in Jilin province, varying from 12.4 to 45.7 μg m$^{-3}$. The contribution of NCP emissions to Shanxi and Shanxi provinces becomes increasingly significant during the episode, with an average of 35.1 μg m$^{-3}$ for Shanxi province and 20.7 μg m$^{-3}$ for Shaanxi province, respectively. The $O_3$ concentration in Inner Mongolia is less influenced by the NCP emissions, with an average of 8.4 μg m$^{-3}$. Figure 12 illustrates the episode-averaged afternoon $O_3$ percentage contribution of the NCP emissions to the surrounding areas. In the NEC, the NCP emissions account for 15-35% of the afternoon $O_3$ concentration in most areas of Liaoning province, and 10-30% in Jilin province. In the NWC, the NCP emissions contribute 10-35% of the $O_3$ concentration in Shanxi province, and 10-25% in Shaanxi. In Inner Mongolia, the impact of the NCP emissions on $O_3$ formation is small, generally less than 15% except in the southern area adjacent to the NCP and Liaoning province where a contribution of more than 10% is found. On average, the NCP emissions distinctly increase the afternoon $O_3$ concentrations in Liaoning, Jilin, Shanxi, Shaanxi, and Inner Mongolia, with the average percentage of 27.4%, 19.5%, 21.2%, 15.8%, and 8.0%, respectively (Table 3).

Additional sensitivity studies have also been performed to examine the potential influences of the cumulus parameterization on evaluation of the contribution of the NCP

**364** emissions to the $PM_{2.5}$ and $O_3$ concentrations in the NEC and NWC, in which the cumulus

**365** parameterization is turned off. The difference of the contribution of NCP emissions to the

**366** $PM_{2.5}$ and $O_3$ concentrations in the NEC and NWC is less than 0.8% between the simulations

**367** with and without the cumulus parameterization. Furthermore, it is worth noting that

**368** uncertainties from meteorological field simulations, emission inventories, and the chemical

**369** mechanism used in simulations, have large potentials to influence evaluation of the effect of

**370** the NCP emissions on the $PM_{2.5}$ and $O_3$ concentrations in the NEC and NWC (Carter and

**371** Atkinson, 1996; Lei et al., 2004; Song et al., 2009; Bei et al., 2017).

**372**

**373** **4    Summary and Conclusions**

**374**    Analyses of the synoptic pattern during the ASM season show that the southeasterly-

**375** southwesterly winds prevail in Northern China, facilitating the trans-boundary transport of air

**376** pollutants from the NCP to the NEC and NWC. The good relationships of $PM_{2.5}$ and $O_3$

**377** concentrations in the NCP with those in the NEC and NWC during the ASM season also

**378** indicate the possibility that the air quality in the NEC and NWC is influenced by the

**379** trans-boundary transport of air pollutants originated from the NCP.

**380**    A widespread and severe pollution episode from 22 to 28 May 2015 in Northern China

**381** is further simulated using the WRF-CHEM model to investigate the impact of trans-boundary

**382** transport of the NCP emissions on $PM_{2.5}$ and $O_3$ concentrations in the NEC and NWC, when

**383** the region is affected by prevailing southeasterly-southwesterly winds associated with the

**384** ASM.

**385**    In general, the WRF-CHEM model well reproduces the temporal variations and spatial

**386** distributions of $PM_{2.5}$, $O_3$, and $NO_2$ concentrations compared to observations in Northern

**387** China, although the model biases still exist due to the uncertainties in simulated

**388** meteorological fields and the emission inventory. The model also performs reasonably well in

simulating the variations of aerosol constituents against the ACSM measurement at the
NCNST site in Beijing.
The FSA method is used to investigate the contribution of trans-boundary transport of
the NCP emissions to $O_3$ and $PM_{2.5}$ levels in the NEC and NWC. Model results show that the
NCP emissions contribute approximately an average of 24.2 and 13.9 μg m$^{-3}$ to the $PM_{2.5}$
concentration in Liaoning and Shanxi during the episode, with the average percentage
contribution of 40.6% and 32.2%, respectively. The NCP emissions enhance the $PM_{2.5}$ level
by 9.6 and 6.5 μg m$^{-3}$ in Jilin and Shaanxi on average, with the percentage contribution of
27.5% and 20.9%, respectively. The NCP emissions also substantially influence the $O_3$
concentration in the NEC and NWC. The NCP emissions increase the afternoon (12:00 -
18:00 BJT) $O_3$ concentration in Liaoning by 46.5 μg m$^{-3}$ on average during the episode,
followed by 35.1 μg m$^{-3}$ in Shanxi, 28.7 μg m$^{-3}$ in Jilin, and 20.7 μg m$^{-3}$ in Shaanxi, with the
average percentage contribution of 27.4%, 21.2%, 19.5%, and 15.8%, respectively. In
contrast, the contribution of trans-boundary transport of the NCP emissions to the $PM_{2.5}$ and
$O_3$ concentration in Inner Mongolia are less, with an average of 2.6 and 8.4 μg m$^{-3}$,
respectively. Our results demonstrate that when southerly winds are prevailing in Northern
China, air pollutants originated from the NCP are likely to be transported northwards and
profoundly affect the air quality in the NEC and NWC. Stringent control of the NCP
emissions not only mitigates the local air pollution, also is beneficial to the air quality in the
NEC and NWC during the ASM season.
It is worth noting that interactions between the air pollution in China and ASM are
two-way and their relationships are complicated and interrelated, especially with regard to the
aerosol-meteorology interaction. Aerosol impacts on meteorology is significant due to its
direct and indirect effects, which further influence the air pollution condition in the lower
troposphere. Aerosol semi-direct effect induced by the light absorbing aerosols in the

**414** atmosphere stabilizes planetary boundary layer (PBL) and thus reduces the PBL height to

**415** exacerbate accumulation of air pollutants within the PBL, particularly for the aging process

**416** of black carbon which considerably enhances light absorption (Wang et al., 2013; Khalizov et

**417** al., 2009; Peng et al., 2016). In addition, aerosol plays an important role in the process of

**418** cloud formation and precipitation via acting as cloud condensation nuclei (CCN) and ice

**419** nuclei (IC). Therefore, aerosol-cloud interactions modify temperature and moisture profiles

**420** and influence precipitation, leading to potential feedback on the atmospheric chemistry

**421** (Wang et al., 2011). In addition, the ASM substantially influence spatial characteristics of the

**422** air pollutants transport and distribution in Eastern China on seasonal, inter-annual, and

**423** decadal scales (Wu et al., 2016). Further studies need to be performed to investigate the

**424** impacts of the ASM variation on the air pollutants transport, which is modulated by climate

**425** changes.

**426** Although the model performs well in simulating $PM_{2.5}$, $O_3$ and $NO_2$ during the episode

**427** in northern China, the uncertainties from meteorological fields and the emission inventory

**428** still exist. Future studies need to be conducted to improve the WRF-CHEM model

**429** simulations, and to further assess the contributions of trans-boundary transport of the NCP

**430** emissions under specific synoptic patterns, considering the rapid changes in anthropogenic

**431** emissions, which is not reflected in the present study. Therefore, more episode simulations

**432** during the ASM season should be performed to comprehensively evaluate the contribution of

**433** trans-boundary transport of the NCP emissions to the air quality in its downwind regions and

**434** support the design and implementation of effective emission control strategies.

**435**


*Acknowledgements*. This work is financially supported by the National Key R&D Plan
(Quantitative Relationship and Regulation Principle between Regional Oxidation Capacity of
Atmospheric and Air Quality (2017YFC0210000)). Naifang Bei is supported by the National
Natural Science Foundation of China (no. 41275101 and no. 41430424) and the Fundamental
Research Funds for the Central Universities of China. Guohui Li is supported by the Hundred
Talents Program of the Chinese Academy of Sciences and the National Natural Science
Foundation of China (no. 41661144020).

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

Contributions of trans-boundary transport to summertime air quality in Beijing, China,
Atmospheric Chemistry and Physics, 17, 1-46, 2017.
Xu, J., Zhang, X. L., Xu, X. F., Zhao, X. J., Meng, W., and Pu, W. W.: Measurement of
surface ozone and its precursors in urban and rural sites in Beijing, in: Second
International Conference on Mining Engineering and Metallurgical Technology, edited
by: Zhu, R., Procedia Earth and Plantetary Science, Elsevier Science Bv, Amsterdam,
668     255-261, 2011.

Zhang, L., Liao, H., and Li, J. P.: Impacts of Asian summer monsoon on seasonal and
interannual variations of aerosols over eastern China, Journal of Geophysical
Research-Atmospheres, 115, 20, 10.1029/2009jd012299, 2010.
Zhang, Q., Streets, D. G., Carmichael, G. R., He, K. B., Huo, H., Kannari, A., Klimont, Z.,
Park, I. S., Reddy, S., Fu, J. S., Chen, D., Duan, L., Lei, Y., Wang, L. T., and Yao, Z. L.:
Asian emissions in 2006 for the NASA INTEX-B mission, Atmospheric Chemistry and
Physics, 9, 5131-5153, 2009.
Zhang, Q. Y., Tao, S. Y., and Chen, L. T.: The inter-annual variability of East Asian summer
monsoon indices and its association with the pattern of general circulation over East Asia
(in Chinese), Acta Meteorological Sinca, 61, 559-568, 2003.
Zhang, R., Khalizova, A. F., Wang, L., Hu, M., and Xu, W.: Nucleation and growth of
nanoparticles in the atmosphere, Chemical Reviews. 112, 1957–2011, 2012.
Zhang, R., Jing, J., Tao, J., Hsu, S. C., Wang, G., Cao, J., Lee, C. S. L., Zhu, L., Chen, Z.,
Zhao, Y., and Shen, Z.: Chemical characterization and source apportionment of $PM_{2.5}$ in
Beijing: seasonal perspective, Atmospheric Chemistry and Physics, 13, 7053-7074,
10.5194/acp-13-7053-2013, 2013.
Zhang, R. Y., Wang, G. H., Guo, S., Zarnora, M. L., Ying, Q., Lin, Y., Wang, W. G., Hu, M.,
and Wang, Y.: Formation of Urban Fine Particulate Matter, Chemical Reviews, 115,
3803-3855, 10.1021/acs.chemrev.5b00067, 2015.
Zhao, J., Levitt, N. P., Zhang, R. Y., and Chen, J. M.: Heterogeneous reactions of
methylglyoxal in acidic media: implications for secondary organic aerosol formation,
Environmental Science and Technology, 40, 7682–7687, 2006.
Zhao, C., Wang, Y. H., Yang, Q., Fu, R., Cunnold, D., and Choi, Y.: Impact of East Asian
summer monsoon on the air quality over China: View from space, Journal of Geophysical
Research-Atmospheres, 115, 12, 10.1029/2009jd012745, 2010.
Zhao, X. J., Zhao, P. S., Xu, J., Meng, W., Pu, W. W., Dong, F., He, D., and Shi, Q. F.:
Analysis of a winter regional haze event and its formation mechanism in the North China
Plain, Atmospheric Chemistry and Physics, 13, 5685-5696, 10.5194/acp-13-5685-2013,
697     2013.

Zhu, B., Akimoto, H., Wang, Z., Sudo, K., Tang, J., and Uno, I.: Why does surface ozone
peak in summertime at Waliguan?, Geophysical Research Letters, 31, 4,
10.1029/2004gl020609, 2004.
Zhu, J. L., Liao, H., and Li, J. P.: Increases in aerosol concentrations over eastern China due
to the decadal-scale weakening of the East Asian summer monsoon, Geophysical
Research Letters, 39, 6, 10.1029/2012gl051428, 2012.
Zhuang, X. L., Wang, Y. S., He, H., Liu, J. G., Wang, X. M., Zhu, T. Y., Ge, M. F., Zhou, J.,
Tang, G. Q., and Ma, J. Z.: Haze insights and mitigation in China: An overview, Journal
of Environmental Sciences, 26, 2-12, 10.1016/s1001-0742(13)60376-9, 2014.




**712**     Table 1 WRF-CHEM model configurations

**713**

| Regions | Northern China |
|---|---|
| Simulation period | May 22 to 28, 2015 |
| Domain size | 350 × 350 |
| Domain center | 35°N, 114°E |
| Horizontal resolution | 10km × 10km |
| Vertical resolution | 35 vertical levels with a stretched vertical grid with spacing ranging from 30 m near the surface, to 500 m at 2.5 km and 1 km above 14 km |
| Microphysics scheme | WSM 6-class graupel scheme (Hong and Lim, 2006) |
| Boundary layer scheme | MYJ TKE scheme (Janjić, 2002) |
| Surface layer scheme | MYJ surface scheme (Janjić, 2002) |
| Cumulus scheme | Kain-Fritsch (new Eta) scheme (Kain, 2004) |
| Land-surface scheme | Unified Noah land-surface model (Chen and Dudhia, 2001) |
| Longwave radiation scheme | Goddard longwave scheme (Chou and Suarez, 2001) |
| Shortwave radiation scheme | Goddard shortwave scheme (Chou and Suarez, 1999) |
| Meteorological boundary and initial conditions | NCEP 1°×1° reanalysis data |
| Chemical initial and boundary conditions | MOZART 6-hour output (Horowitz et al., 2003) |
| Anthropogenic emission inventory | SAPRC-99 chemical mechanism emissions (Zhang et al., 2009) |
| Biogenic emission inventory | MEGAN model developed by Guenther et al. (2006) |
| Model spin-up time | 28 hours |

**714**

**715**

**716**

Table 2 Daily average PM$_{2.5}$ contributions (µg m$^{-3}$) of the NCP emissions in the NEC and NWC from 22 to 28 May 2015.

| Date | Jilin | Liaoning | Shanxi | Shaanxi | Inner Mongolia |
|---|---|---|---|---|---|
| 22 | 0.7±0.4 | 6.1±4.5 | 0.7±1.1 | 0.1±0.0 | 0.2±0.1 |
| 23 | 6.1±2.1 | 15.4±4.6 | 4.7±5.5 | 0.5±0.3 | 1.0±0.5 |
| 24 | 10.0±2.1 | 19.6±5.8 | 12.7±5.0 | 3.5±1.6 | 2.2±1.3 |
| 25 | 14.4±4.4 | 33.6±8.9 | 14.6±5.1 | 6.0±1.7 | 2.6±1.8 |
| 26 | 6.4±2.8 | 24.1±9.5 | 16.3±5.7 | 9.1±1.8 | 1.9±0.8 |
| 27 | 11.4±3.6 | 46.7±12.3 | 20.7±7.3 | 11.6±2.2 | 3.2±1.9 |
| 28 | 18.0±7.4 | 23.7±8.5 | 27.5±9.0 | 14.9±4.4 | 6.9±3.4 |
| Average (µg m$^{-3}$) | 9.6±3.3 | 24.2±7.7 | 13.9±5.5 | 6.5±1.7 | 2.6±1.4 |
| Average (%) | 27.5±7.8 | 40.6±9.7 | 32.2±9.4 | 20.9±4.1 | 16.7±6.5 |

Table 3 Daily afternoon (12:00-18:00 BJT) average $O_3$ contributions ($\mu$g m$^{-3}$) of the NCP emissions in the NEC and NWC from 22 to 28 May 2015.

| Date | Jilin | Liaoning | Shanxi | Shaanxi | Inner Mongolia |
|---|---|---|---|---|---|
| 22 | 12.4±0.1 | 23.9±2.7 | 12.7±0.0 | 7.7±0.0 | 2.8±0.0 |
| 23 | 25.8±2.5 | 38.9±6.2 | 21.5±1.1 | 13.1±0.3 | 5.1±0.2 |
| 24 | 35.0±3.6 | 47.5±8.1 | 31.3±3.9 | 21.2±1.9 | 8.5±0.5 |
| 25 | 45.7±8.4 | 69.5±15.5 | 39.7±6.4 | 21.5±2.5 | 9.9±0.6 |
| 26 | 16.6±1.6 | 41.0±5.9 | 36.4±4.6 | 21.7±2.4 | 10.8±0.7 |
| 27 | 23.9±5.0 | 69.3±16.4 | 51.7±7.8 | 33.5±4.5 | 9.6±1.0 |
| 28 | 41.7±5.5 | 35.1±6.5 | 52.3±9.0 | 26.5±4.7 | 12.2±1.8 |
| Average ($\mu$g m$^{-3}$) | 28.7±3.8 | 46.5±8.8 | 35.1±4.7 | 20.7±2.3 | 8.4±0.7 |
| Average (%) | 19.5±2.9 | 27.4±5.9 | 21.2±3.2 | 15.8±2.0 | 8.0±0.7 |

Figure 1 WRF-CHEM simulation domain with topography. The blue circles represent centers of cities with ambient monitoring sites and the red circle denotes the NCNST site. The size of the blue circle represents the number of ambient monitoring sites of cities.

Figure 2 (a) Geopotential heights and (b) the mean sea level pressures with wind vectors during the summer monsoon season in 2015.

Figure 3 Relationships of observed $PM_{2.5}$ and $O_3$ concentrations in NCP with those in the NEC during May to September from 2013 to 2016.

Figure 4 Same as Figure 3, but for the NWC.

Figure 5 Comparison of measured (black dots) and predicted (blue line) diurnal profiles of near-surface hourly (a) $PM_{2.5}$, (b) $O_3$, and (c) $NO_2$ averaged over all ambient monitoring stations in Northern China from 22 to 28 May 2015.

Figure 6 Comparison of measured (black dots) and simulated (black line) diurnal profiles of submicron aerosol species of (a) POA, (b) SOA, (c) sulfate, (d) nitrate, and (e) ammonium at NCNST site in Beijing from 22 to 28 May 2015.

Figure 7 Pattern comparison of simulated vs. observed near-surface $PM_{2.5}$ at 08:00 BJT during from 23 to 28 May 2015. Colored circles: $PM_{2.5}$ observations; color contour: $PM_{2.5}$ simulations; black arrows: simulated surface winds.

Figure 8 Same as Figure 7, but for the near-surface $O_3$ at 14:00 BJT.

Figure 9 Contributions of NCP emissions to the daily mean near-surface $PM_{2.5}$ concentration in the NEC and NWC from 23 to 28 May 2015.

Figure 10 Average percentage contribution of NCP emissions to $PM_{2.5}$ concentrations in the NEC and NWC from 22 to 28 May 2015.

Figure 11 Same as Figure 9, but for the afternoon (12-18:00 BJT) $O_3$ concentration.

Figure 12 Same as Figure 10, but for the afternoon (12-18:00 BJT) $O_3$ concentration.

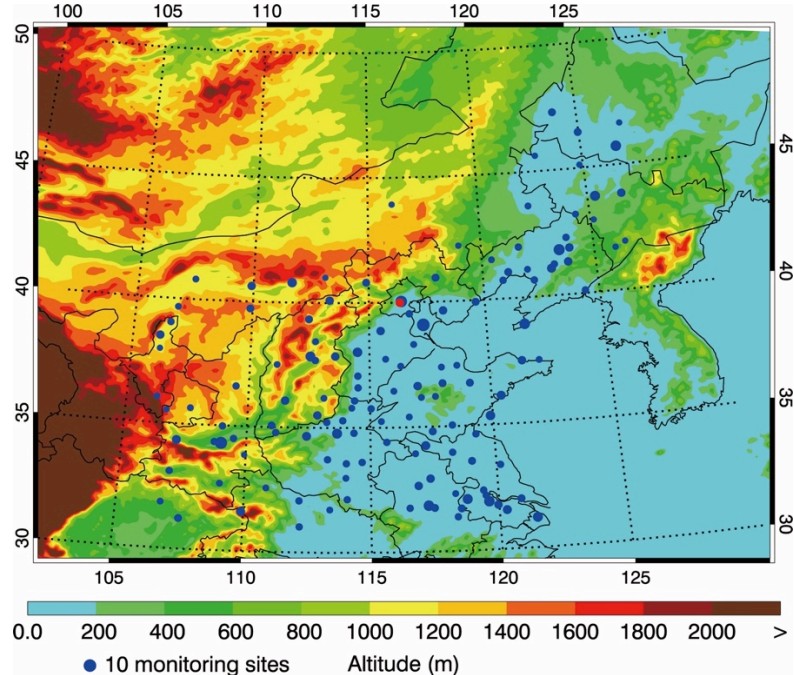



Figure 1 WRF-CHEM simulation domain with topography. The blue circles represent centers
of cities with ambient monitoring sites and the red circle denotes the NCNST site. The size of
the blue circle represents the number of ambient monitoring sites of cities.





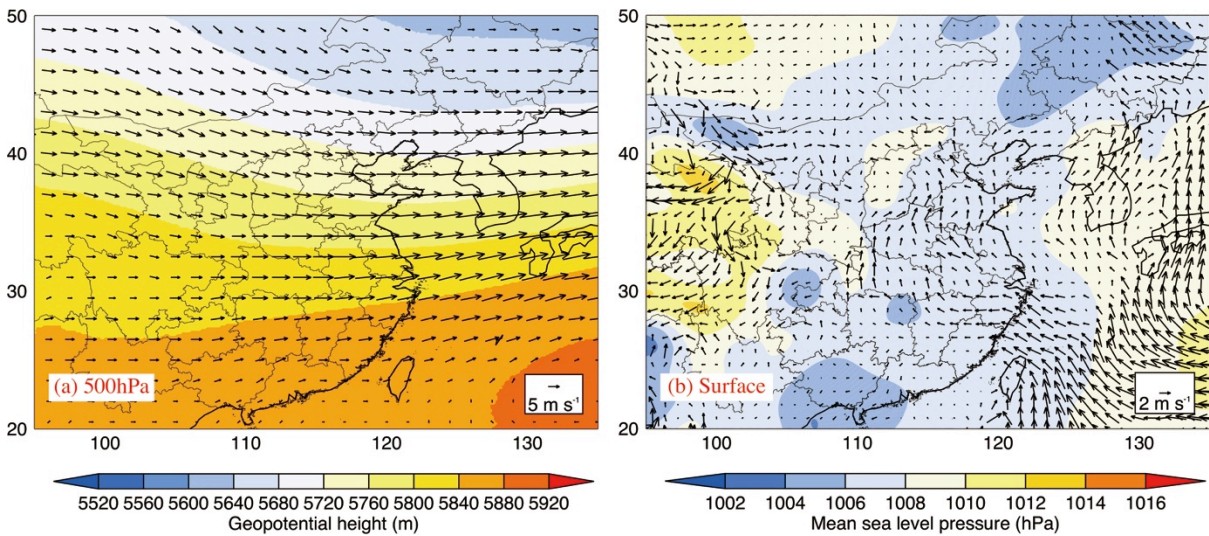



Figure 2 (a) Geopotential heights and (b) the mean sea level pressures with wind vectors
during the summer monsoon season in 2015.






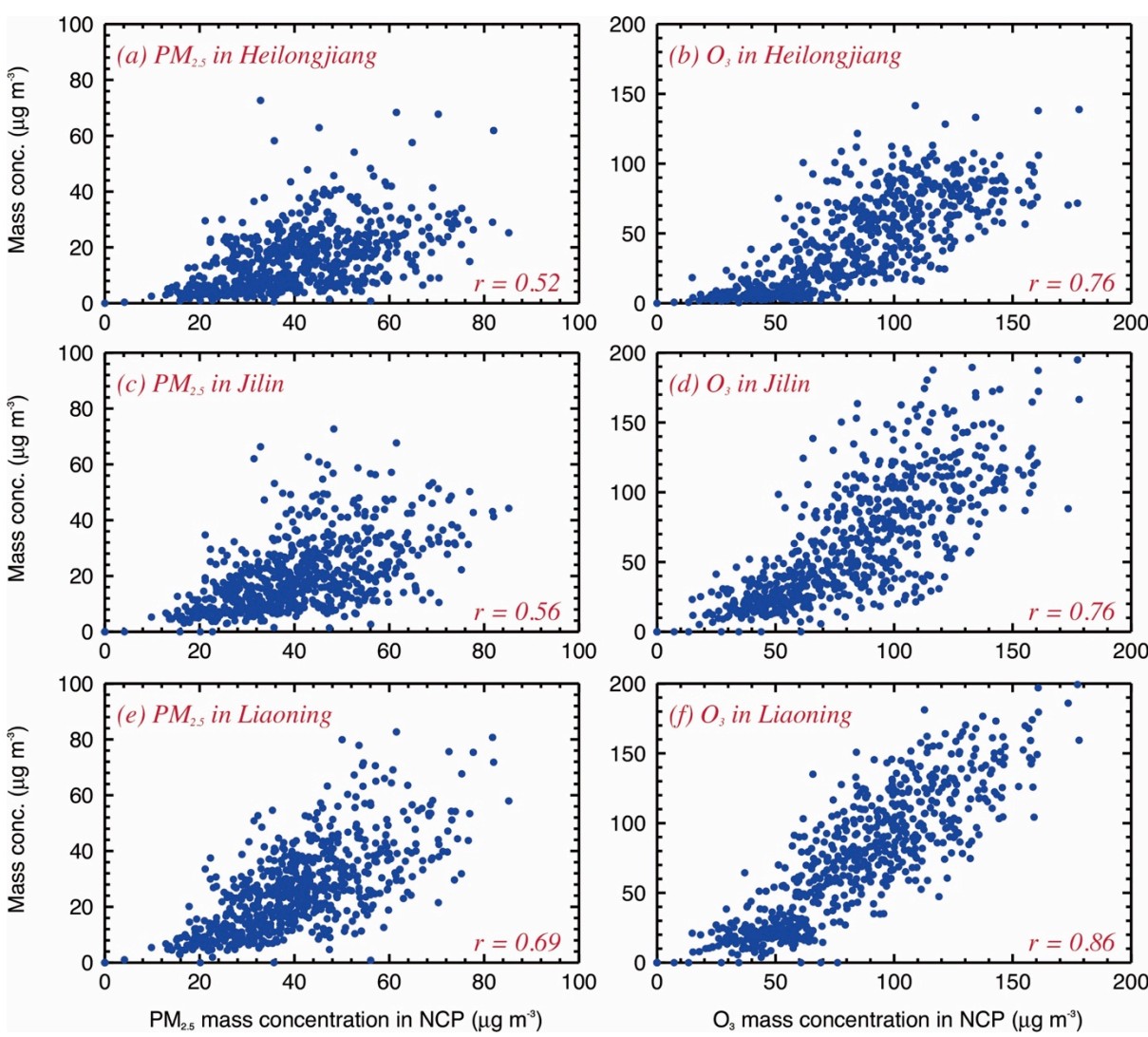

Figure 3 Relationships of observed $PM_{2.5}$ and $O_3$ concentrations in NCP with those in the NEC during May to September from 2013 to 2016.

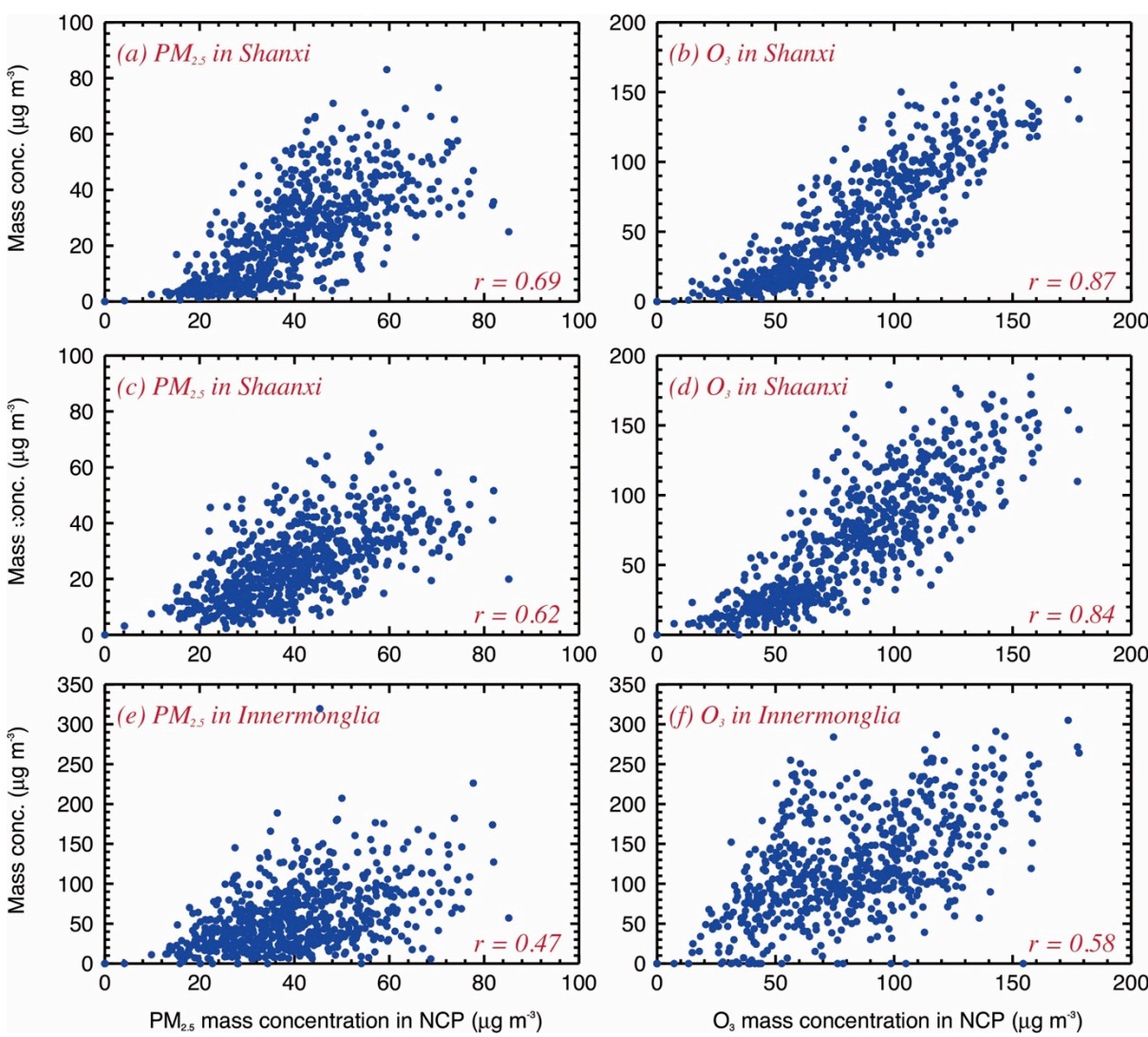

Figure 4 Same as Figure 3, but for the NWC.

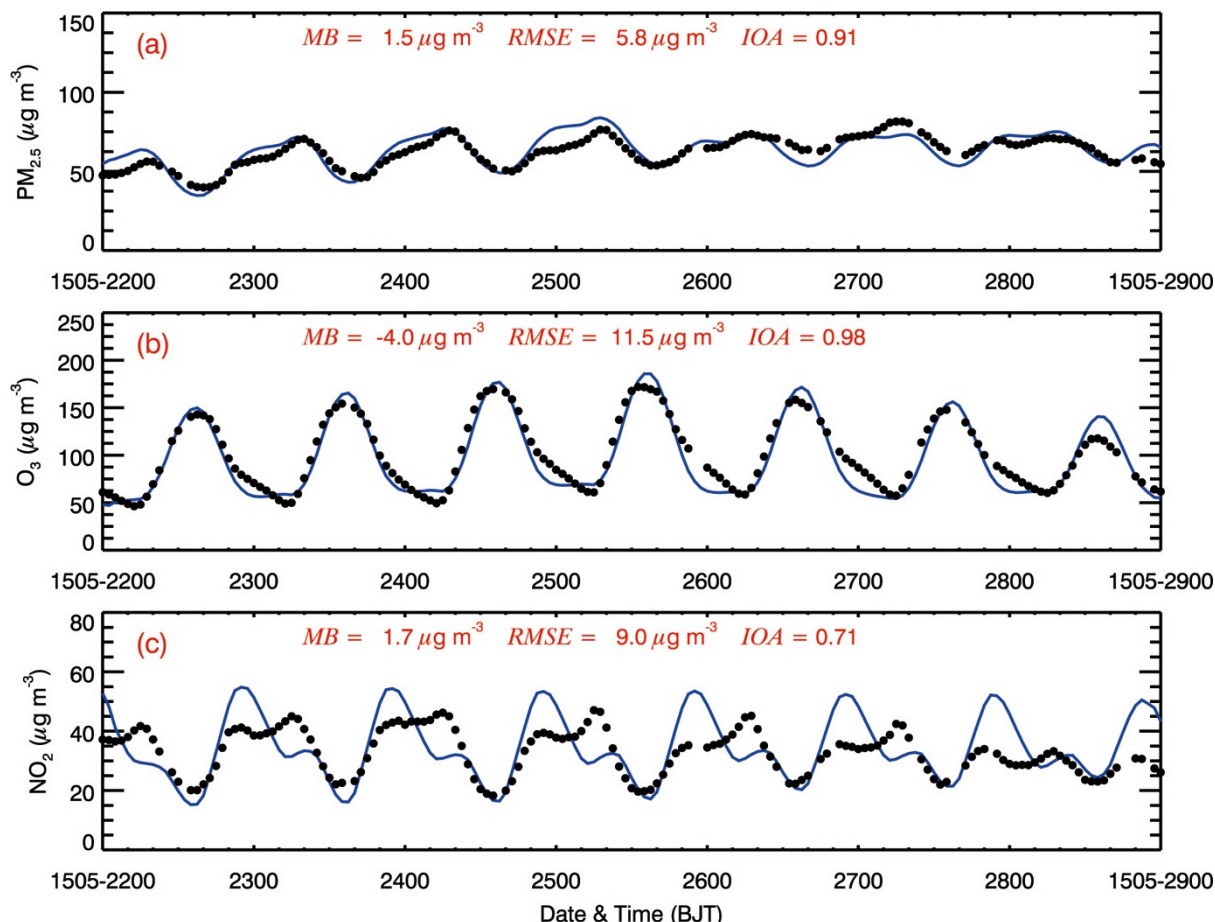



Figure 5 Comparison of measured (black dots) and predicted (blue line) diurnal profiles of
near-surface hourly (a) $PM_{2.5}$, (b) $O_3$, and (c) $NO_2$ averaged over all ambient monitoring
stations in Northern China from 22 to 28 May 2015





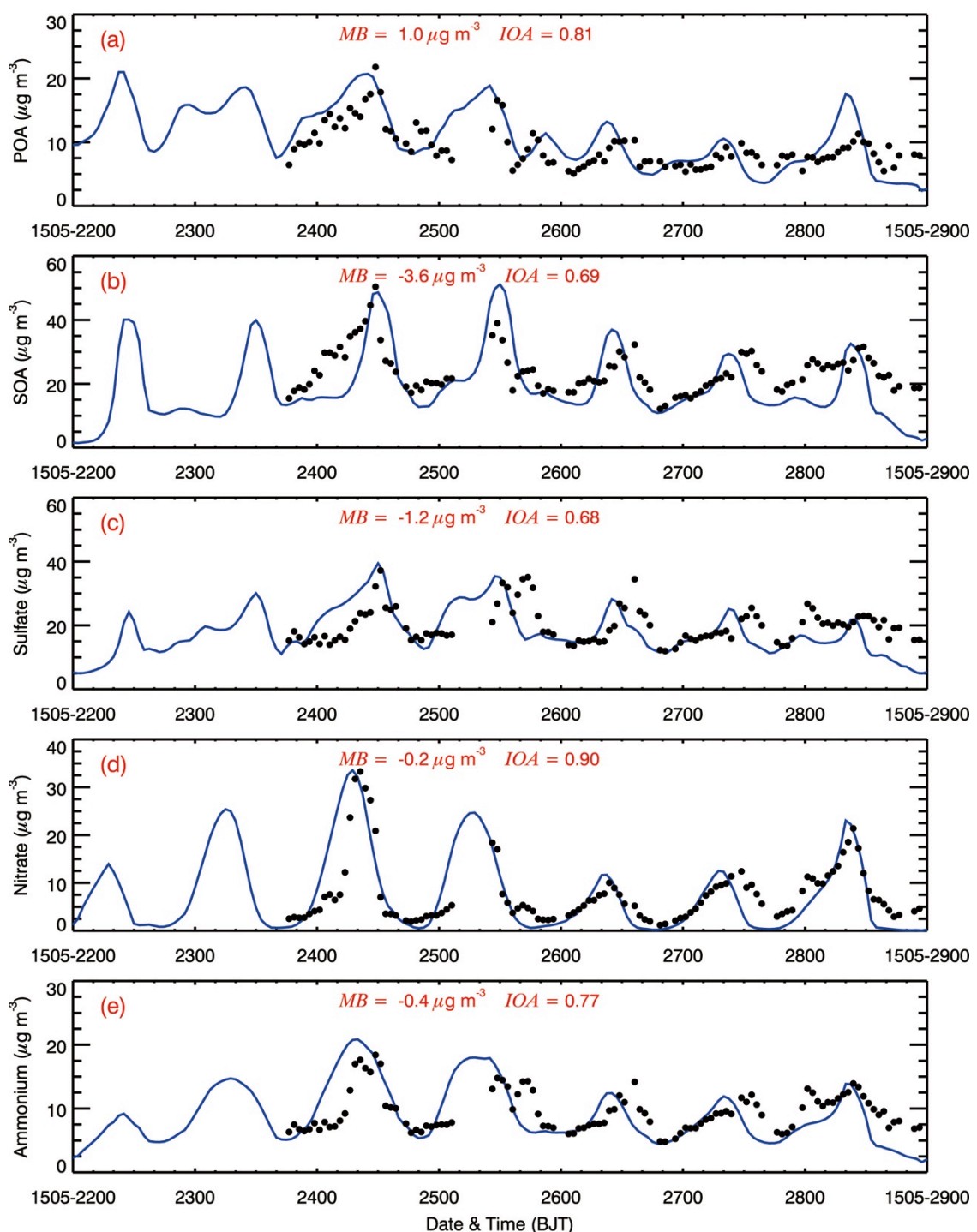

**809**

**810**

**811** Figure 6 Comparison of measured (black dots) and simulated (black line) diurnal profiles of
**812** submicron aerosol species of (a) POA, (b) SOA, (c) sulfate, (d) nitrate, and (e) ammonium at
**813** NCNST site in Beijing from 22 to 28 May 2015.

**814**

**815**

**816**

**817**

**818**

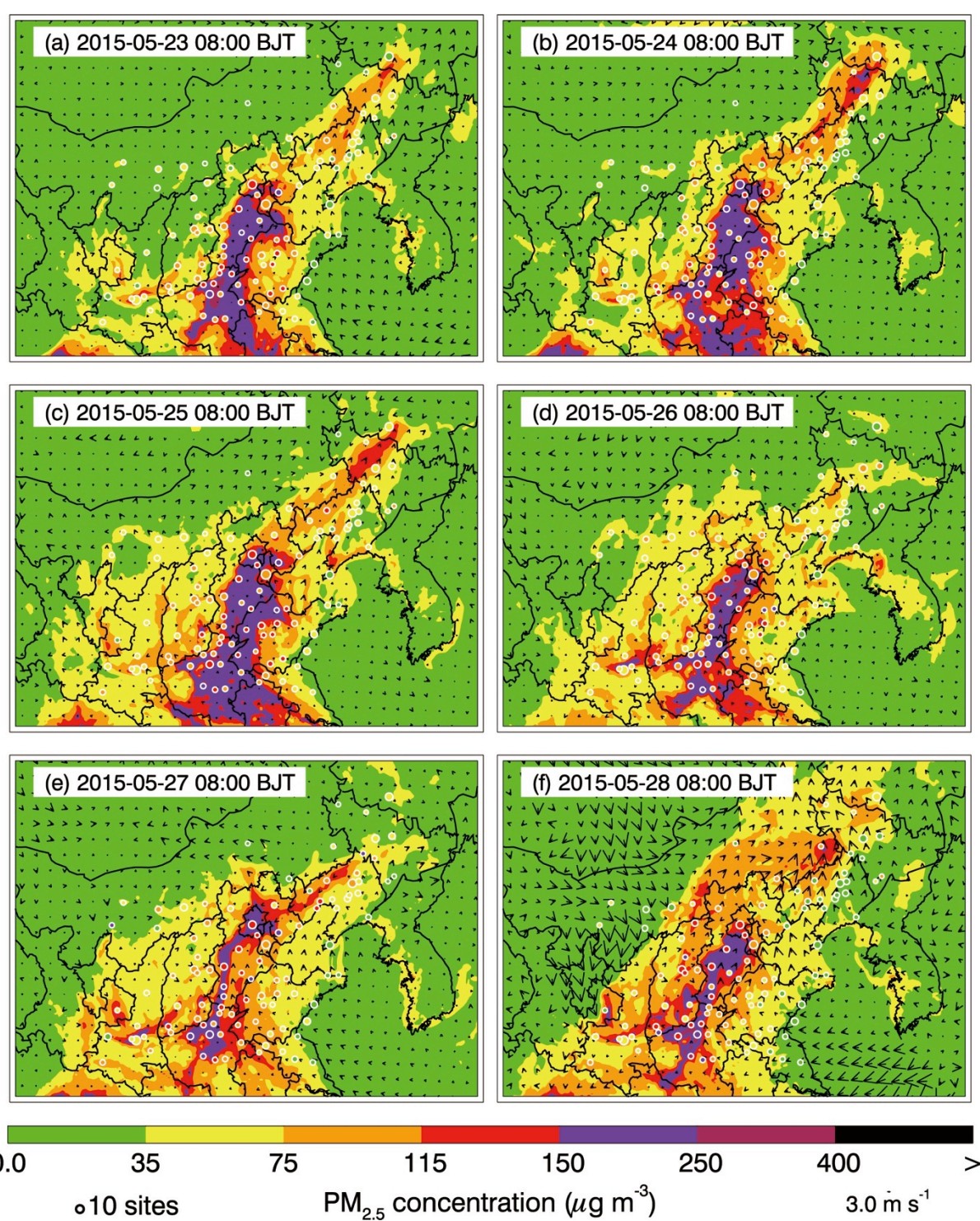

Figure 7 Pattern comparison of simulated vs. observed near-surface PM$_{2.5}$ at 08:00 BJT during from 23 to 28 May 2015. Colored circles: PM$_{2.5}$ observations; color contour: PM$_{2.5}$ simulations; black arrows: simulated surface winds.

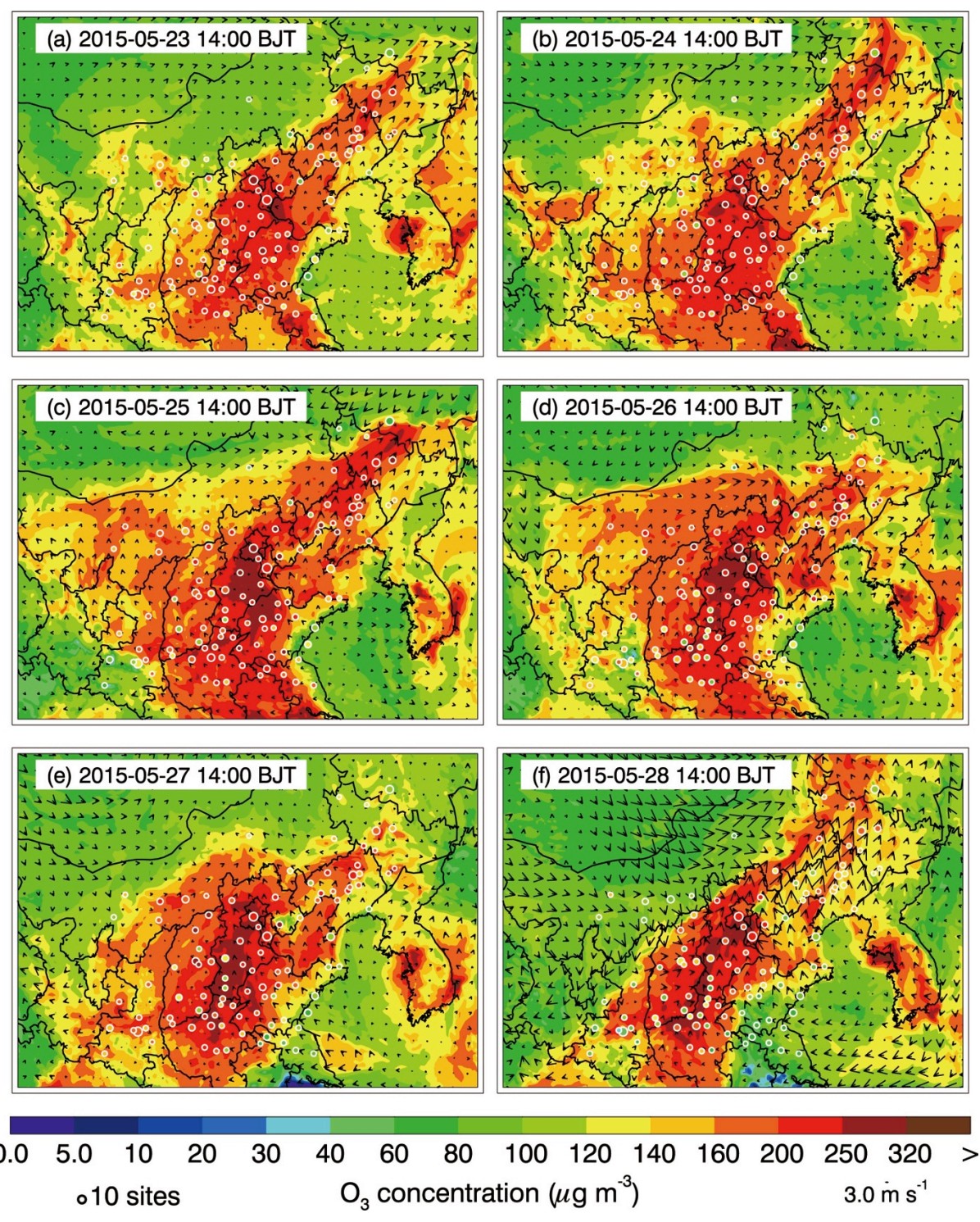

Figure 8 Same as Figure 7, but for the near-surface O$_3$ at 14:00 BJT.

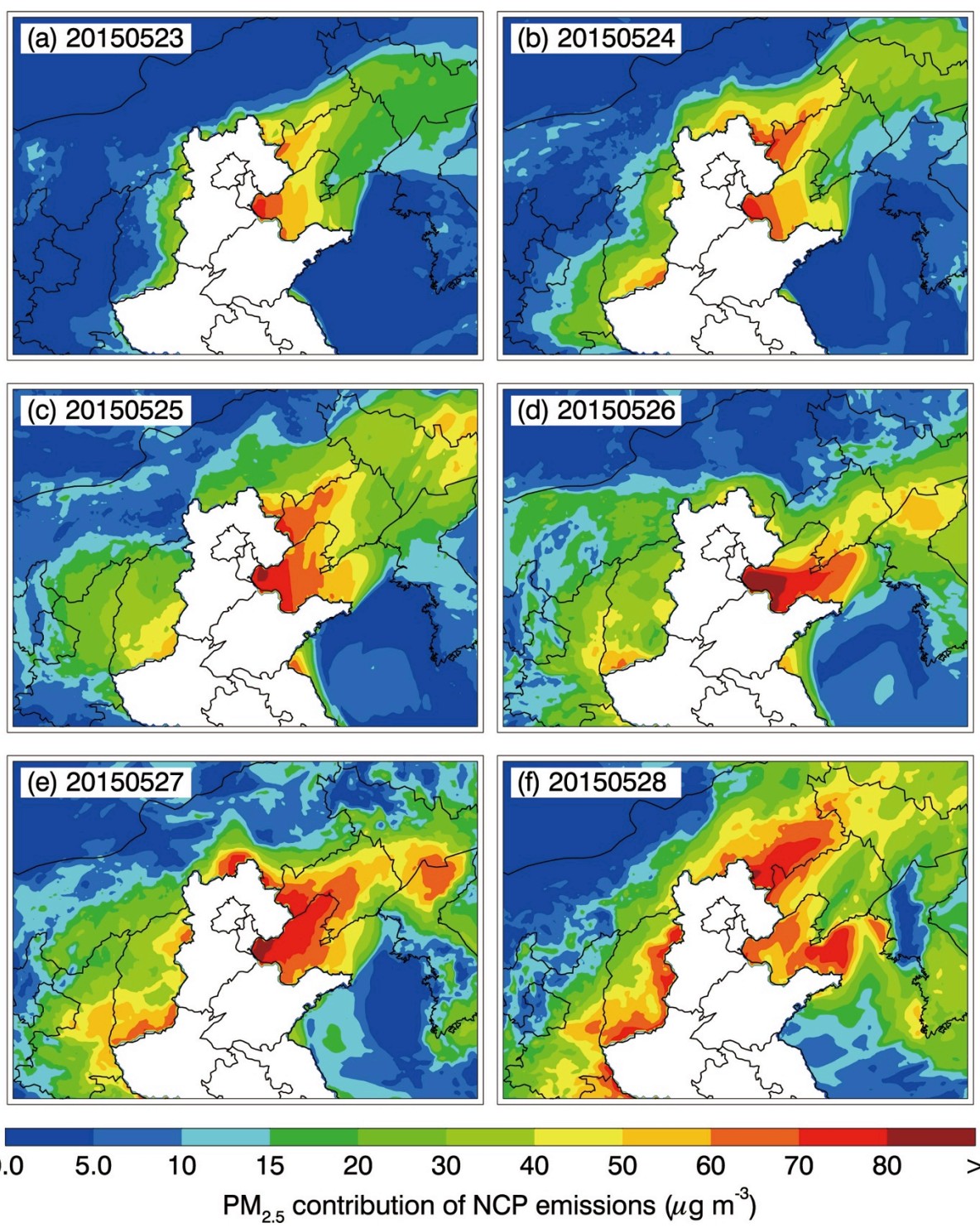

Figure 9 Contributions of NCP emissions to the daily mean near-surface PM$_{2.5}$ concentration
in the NEC and NWC from 23 to 28 May 2015.

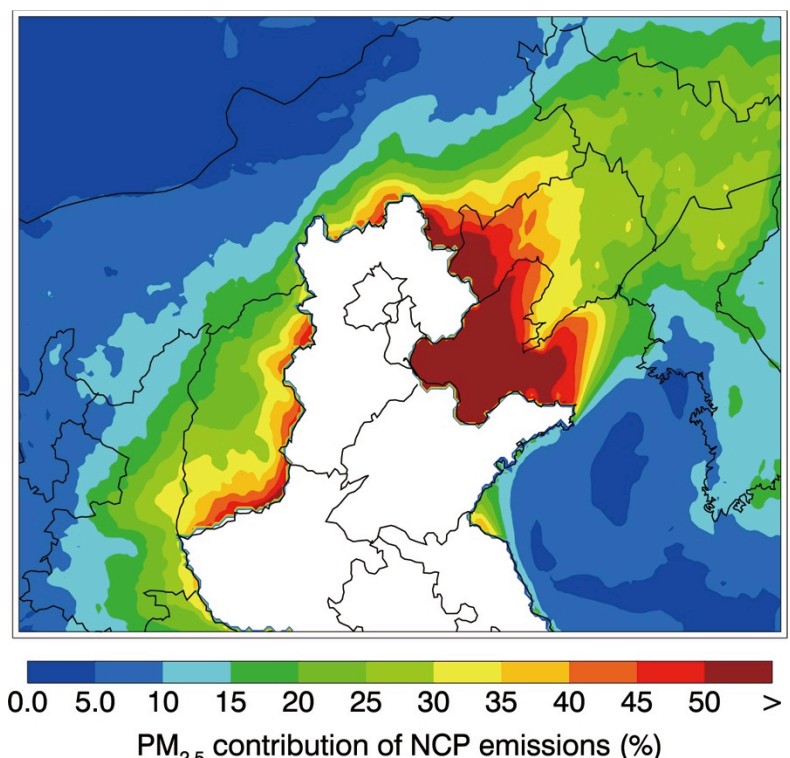

Figure 10 Average percentage contribution of NCP emissions to PM$_{2.5}$ concentrations in the
NEC and NWC fron 22 to 28 May 2015.

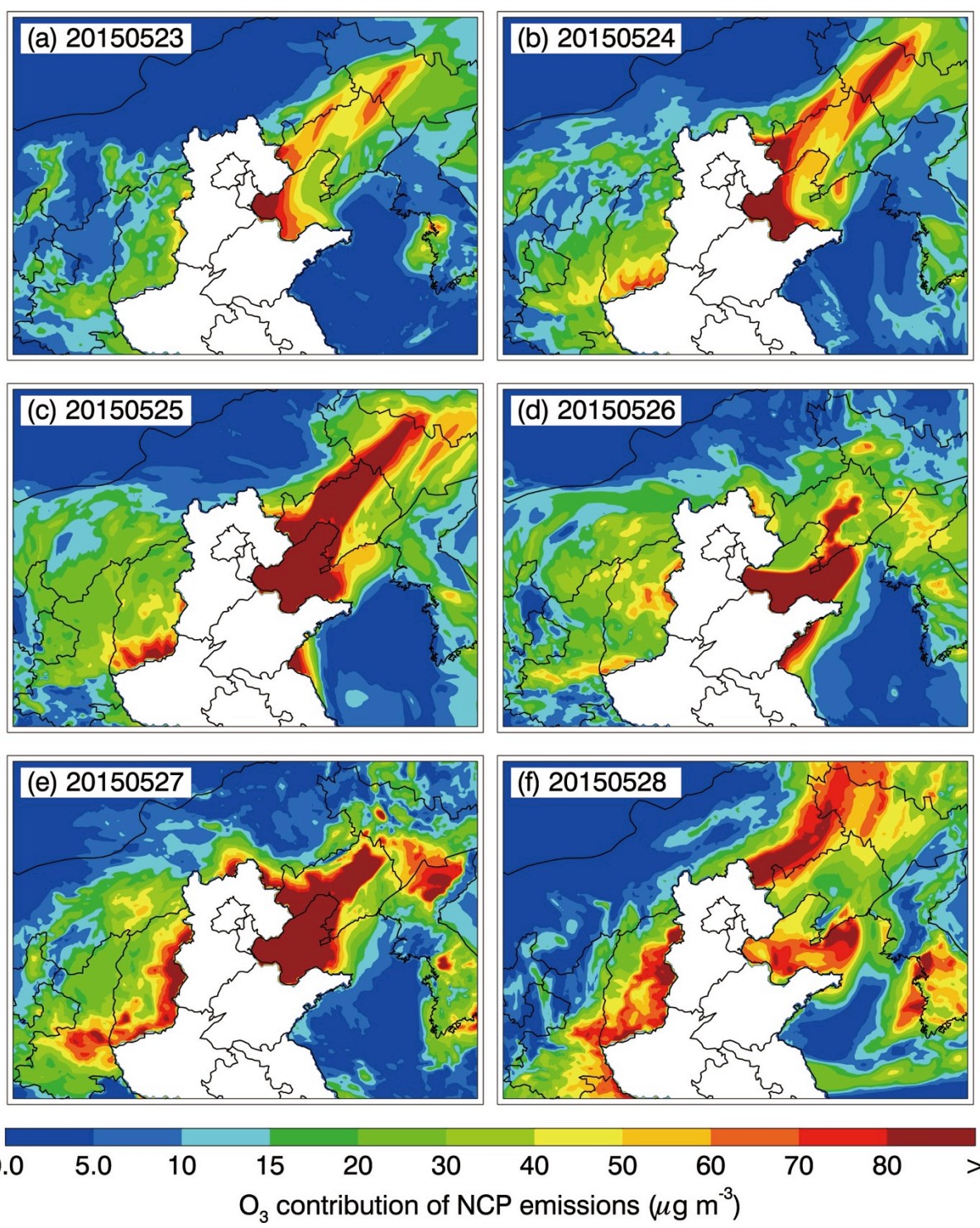

Figure 11 Same as Figure 9, but for the afternoon (12-18:00 BJT) O₃ concentration.

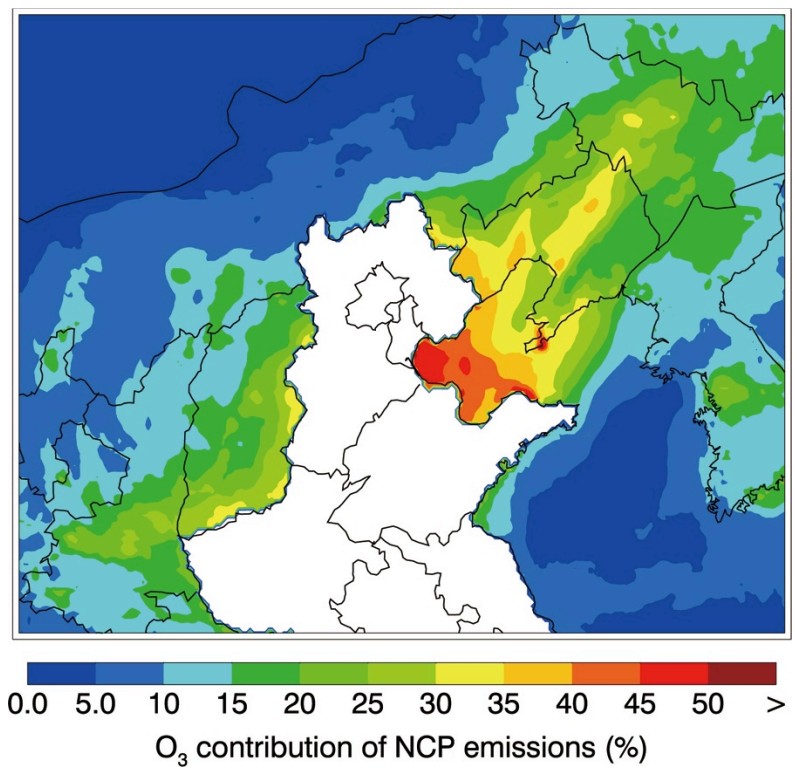

864    Figure 12 Same as Figure 10, but for the afternoon (12-18:00 BJT) $O_3$ concentration.