# Peer review of "Widespread air pollutants of the North China Plain during the Asian summer monsoon season: A case study"

_Atmospheric Chemistry and Physics, 2017_

## Referee Comment (RC1) · Anonymous Referee #1 · 25 Jan 2018

This manuscript presents an investigation of the impacts of trans-boundary transport of air pollutants originated from the North China Plain on regional air quality in the Northeast and Northwest China. Contributions of air pollutants from neighboring regions to local air quality become significant especially under prevailing meteorological conditions such as Asian Summer Monsoon seasons (ASM). However, it is quite difficult to assess to what extents the impacts of the trans-boundary transport are and to date few studies are available in the literature, hindering the effective measures from proposing regarding pollution control and prevention. The paper is well written and organized and only a few minor issues need to be resolved before its final publication in the journal. 1. When quantitatively assessing the impacts of the trans-boundary pollutants from NCP

on the NEC or NWC regions, it is necessary to estimate the uncertainties and include them in the evaluation. In addition, a clear list of all possible sources of uncertainties is needed in the assessment. 2. The validation of separating different contributions (e.g., local, transport etc.) seems to be lack of clear support. Why other processes for example secondary reactions of air pollutants are not included in the method of separation? 3. Why PM10 is not included in quantitative evaluations? Is it because not significant in term of concentration or there are other reasons? 4. A few other rather minor points: 1) L32 on p1, however might be better to be moved to the beginning of the sentence. 2) L52 on p2, pollutants emissions? Emissions of pollutants might be better. There are quite a few on other pages. 3) L73 on p3, tend should be tends. 4) L114 on p4 and other pages, "The further description", here "The" is not needed. 5) L152 on p6, it is "Results and Discussion". 6) L190-191 on p8, the values of 0.69 and 0.62 are not significant different, similar for the values of 0.87 and 0.84. 7) There are a few acronyms (i.e., IOA, MB) that needed to be specified. 8) Why 8:00 and 14:00 are respectively used in Figures 7 and 8? Why not other times? 9) L266 on p11, you cannot use something like "the most remarkable". 10) L288 on p11, in most areas not in the most areas.

---

## Referee Comment (RC2) · Anonymous Referee #2 · 4 Apr 2018

This study evaluates the influences of air pollution from North China Plain (NCP) on its surrounding regions, including Northeast and Northwest China (NEC and NWC), when Asian summer monsoon (ASM) is present. The case study with WRF-Chem modeling in this study suggests that the air pollution emitted or formed over NCP could significantly deteriorate the air quality at certain areas in NEC and NWC, particularly in terms of PM2.5 and ozone concentrations. Since the transboundary transport is a key issue in regional air pollution control in China and there is lack of such studies, I recommend publishing this work, after the authors have sufficiently addressed following issues. Major points: 1. The horizontal grid spacing for the simulations in this study is 10 km, which is the lower bound for the WRF model to turn on the cumulus scheme to consider the sub-grid-scale effect of convective and/or shallow clouds. Was any certain cumulus scheme used in this study? Which one was used? If the simulations conducted without the cumulus parameterization, what are the potential influences on the results? 2. Typically, the accuracy in chemical transport model simulations depends on emission inventory, meteorology, and chemistry. The key features in the aerosol chemistry in China are related to very efficient secondary formation (Guo et al., Proc. Natl. Acad. Sci. USA 111, 17373, 2014; Zhang et al., Chem. Rev. 115, 3803, 2015). Specifically, the efficient secondary aerosol processes include aerosol nucleation and rapid growth under favorable conditions (Zhang et al., Chem. Rev. 112, 1957, 2012; Qiu et al., Phys. Chem. Chem. Phys. 15, 5738, 2013). It would be necessary that you clearly state how those processes were accounted for in your chemistry module. 3. Also, aerosol impacts on meteorological fields could be significant, which might further affect the aerosol pollution condition in the lower troposphere. Also, aerosol-cloud interactions might modify temperature and moisture profiles and precipitation (Wang et al., Atmos. Chem. Phys. 11, 12421, 2011), leading to potential feedback on the atmospheric chemistry. Aerosol radiative effects induced by black carbon (BC) or other aerosol components could stabilize boundary layer and thus reduce the height of boundary layer, tending to exacerbate aerosol pollution near ground (Wang et al., Atmos. Environ. 81, 713, 2013). A particular important aspect is the aging of BC, which considerably enhances light absorption (Khalizov et al., J. Phys. Chem. 113, 1066, 2009; Peng et al., Proc. Natl. Acad. Sci. USA 113, 4266, 2016). 4. It would also be necessary to mention the potential impacts of climate changes on pollution conditions in China (Wu et al., Sci. China: Earth Sci. 59, 1–16, 2016). Minor points: As indicated in lines 81-84, the impacts of ASM on the air pollution over Northern China varies with the intensity of ASM. A case study on one-year monsoon season (May 2015) as reported in this work may not represent the various response under different ASM conditions. In addition to carry on more ASM episodes in future, how strong is the ASM season in this work relative to other years and/or the normal situation? A more detailed description of the strength of the simulated ASM will help us to evaluate the

uncertainty range of the results in this study. It is good that the authors discuss the relative contribution of North China Plain to its surrounding regions in Figures 10 and 12, could the authors also provide the mean values of the contributions percentages in Tables 2 and 3? Also, please state the contribution percentages in the abstract. If available, could the authors add the uncertainty in the two tables and discuss it in the body text? In lines 221-222, the work by Wang et al. (PNAS, 20016) is relevant, which has documented the possible efficient SO2 conversion pathway with assistant of NO2 in aqueous phase. Regarding the uncertainties from meteorological fields as mentioned in line 335, how do the simulations perform in predicting the regular meteorological parameters, such as temperature, wind speed, and so on, comparing to observations? In section 3.3 lines 259-260, the authors mentioned that the simulations can be used for evaluating the interactions of the two emissions (i.e., with NCP emissions only and with non-NCP emissions only), but there are no discussions about the interactions in the remaining part of the manuscript. It is interesting to know how possible the non-NCP emissions affect NCP. Could the authors show some results about the interactions of the emissions from the two regions?

---

## Author Comment (AC1) · 16 May 2018

**Reply to Anonymous Referee #1**

We thank the reviewer for the careful reading of the manuscript and helpful comments. We have revised the manuscript following the suggestion, as described below.

This manuscript presents an investigation of the impacts of trans-boundary transport of air pollutants originated from the North China Plain on regional air quality in the Northeast and Northwest China. Contributions of air pollutants from neighboring regions to local air quality become significant especially under prevailing meteorological conditions such as Asian Summer Monsoon seasons (ASM). However, it is quite difficult to assess to what extents the impacts of the trans-boundary transport are and to date few studies are available in the literature, hindering the effective measures from proposing regarding pollution control and prevention. The paper is well written and organized and only a few minor issues need to be resolved before its final publication in the journal.

**1 Comment**: When quantitatively assessing the impacts of the trans-boundary pollutants from NCP on the NEC or NWC regions, it is necessary to estimate the uncertainties and include them in the evaluation. In addition, a clear list of all possible sources of uncertainties is needed in the assessment.

**Response:** We have included the uncertainties (standard deviation) in Table 2 and Table 3 and also clarified in Section 3.3.2: "*Furthermore, it is worth noting that uncertainties from meteorological field simulations, emission inventories, and the chemical mechanism used in simulations, have large potentials to influence evaluation of the effect of the NCP emissions on the $PM_{2.5}$ and $O_3$ concentrations in the NEC and NWC (Carter and Atkinson, 1996; Lei et al., 2004; Song et al., 2009; Bei et al., 2017).*"

**2 Comment:** The validation of separating different contributions (e.g., local, transport etc.) seems to be lack of clear support. Why other processes for example secondary reactions of air pollutants are not included in the method of separation?

**Response**: We have clarified in Supplement Information (SI) Section 2.1: "*The formation of the secondary atmospheric pollutant, such as $O_3$, secondary organic aerosol, and nitrate, is a complicated nonlinear process in which its precursors from various emission sources and*

*transport react chemically or reach equilibrium thermodynamically. Nevertheless, it is not straightforward to evaluate the contributions from different factors in a nonlinear process. The factor separation approach (FSA) proposed by Stein and Alpert (1993) can be used to isolate the effect of one single factor from a nonlinear process and has been widely used to evaluate source effects (Gabusi et al., 2008; Weinroth et al., 2008; Carnevale et al., 2010; Li et al., 2014a).*" The detailed description of factor separation approach can be found in Supplement Information (SI) Section 2.1.

**3 Comment:** Why $PM_{10}$ is not included in quantitative evaluations? Is it because not significant in term of concentration or there are other reasons?

**Response:** We have clarified in Section 3.3: "*In the present study, the effect of the NCP emissions on the $PM_{2.5}$ and $O_3$ concentrations in the NEC and NWC is evaluated, considering that they have the long lifetime of several days in the troposphere and often constitute the primary air pollutant during summertime (Seinfeld and Pandis, 2006). However, the trans-boundary transport of $PM_{10}$ is not considered due to its short lifetime of several hours caused by the dry deposition and gravity and the fact that $PM_{10}$ is generally confined to its source region when the wind is not strong enough (Sun et al., 2006).*"

**A few other rather minor points:**

**1)** L32 on p1, however might be better to be moved to the beginning of the sentence.

**Response:** We have moved "however" to the beginning of the sentence in abstract.

**2)** L52 on p2, pollutants emissions? Emissions of pollutants might be better. There are quite a few on other pages.

**Response:** We have revised "pollutants emissions" as "emissions of pollutants" in Section 1.

**3)** L73 on p3, tend should be tends.

**Response:** We have revised "tend" as "tends" in Section 1.

**4)** L114 on p4 and other pages, "The further description", here "The" is not needed.

**Response:** We have deleted "The" in Section 2.1.

**5)** L152 on p6, it is "Results and Discussion".

**Response:** We have revised "Results and Discussions" as "Results and Discussion" in Section 3.

**6)** L190-191 on p8, the values of 0.69 and 0.62 are not significant different, similar for the values of 0.87 and 0.84.

**Response:** We have clarified in Section 3.1: "*The decreasing trend of the correlation coefficients also exists from east to west in the NWC, with coefficients of 0.69 and 0.62 for PM$_{2.5}$, and 0.87 and 0.84 for O$_3$ in Shanxi and Shaanxi, respectively.*".

**7)** There are a few acronyms (i.e., IOA, MB) that needed to be specified.

**Response:** We have clarified in Section 2.2: "*The mean bias (MB), root mean square error (RMSE) and the index of agreement (IOA) are utilized to evaluate the performance of the WRF-CHEM model simulations against measurements.*" The detailed description about the statistical methods can be found in Supplementary Information (SI).

**8)** Why 8:00 and 14:00 are respectively used in Figures 7 and 8? Why not other times?

**Response:** We have clarified in Section 3.2.3: "*The peak PM$_{2.5}$ concentration generally occurs from 08:00 to 10:00 Beijing Time (BJT) during the simulated episode.*" and "*The O$_3$ concentration during summertime generally reaches its peak from 14:00 to 16:00 BJT in Northern China (Figure 5).*".

**9)** L266 on p11, you cannot use something like "the most remarkable".

**Response:** We have revised "the most remarkable" as "remarkable" in Section 3.3.1.

**10)** L288 on p11, in most areas not in the most areas.

**Response:** We have revised "in the most areas" as "in most areas" in Section 3.3.1.

*ning, Shanxi, Shaanxi and Inner Mongolia, respectively.*".

---

## Author Comment (AC2) · 16 May 2018

**Reply to Anonymous Referee #2**

We thank the reviewer for the careful reading of the manuscript and helpful comments. We have revised the manuscript following the suggestion, as described below.

This study evaluates the influences of air pollution from North China Plain (NCP) on its surrounding regions, including Northeast and Northwest China (NEC and NWC), when Asian summer monsoon (ASM) is present. The case study with WRF-CHEM modeling in this study suggests that the air pollution emitted or formed over NCP could significantly deteriorate the air quality at certain areas in NEC and NWC, particularly in terms of $PM_{2.5}$ and ozone concentrations. Since the trans-boundary transport is a key issue in regional air pollution control in China and there is lack of such studies, I recommend publishing this work, after the authors have sufficiently addressed following issues.

**Major points:**

**1 Comment**: The horizontal grid spacing for the simulations in this study is 10 km, which is the lower bound for the WRF model to turn on the cumulus scheme to consider the sub-grid-scale effect of convective and/or shallow clouds. Was any certain cumulus scheme used in this study? Which one was used? If the simulations conducted without the cumulus parameterization, what are the potential influences on the results?

**Response:** We have clarified in Section 2.1: "*It is worth noting that the horizontal resolution of 10 km adopted in this study is the lower bound for the WRF model to turn on the cumulus scheme, so the new Kain-Fritch scheme is used in the present study (Table 1).*"

We have also clarified in Section 3.3.: "*Additional sensitivity studies have also been performed to examine the potential influences of the cumulus parameterization on evaluation of the contribution of the NCP emissions to the $PM_{2.5}$ and $O_3$ concentrations in the NEC and NWC, in which the cumulus parameterization is turned off. The difference of the contribution of NCP emissions to the $PM_{2.5}$ and $O_3$ concentrations in the NEC and NWC is less than 0.8% between the simulations with and without the cumulus parameterization.*"

**2 Comment:** Typically, the accuracy in chemical transport model simulations depends on emission inventory, meteorology, and chemistry. The key features in the aerosol chemistry in China are related to very efficient secondary formation (Guo et al., Proc. Natl. Acad. Sci. USA 111, 17373, 2014; Zhang et al., Chem. Rev. 115, 3803, 2015). Specifically, the efficient secondary aerosol processes include aerosol nucleation and rapid growth under favorable conditions (Zhang et al., Chem. Rev. 112, 1957, 2012; Qiu et al., Atmos. Chem. Phys. 15, 5738, 2013). It would be necessary that you clearly state how those processes were accounted for in your chemistry module.

**Response:** We have clarified in Section 2.1: "*The key characteristics of the aerosol pollution in China are frequently associated with rather efficient secondary formation, including aerosol nucleation and rapid growth under favorable conditions (Zhang et al., 2012; Qiu et al., 2013; Guo et al., 2014; Zhang et al., 2015). The new particle production rate in the WRF-CHEM model is calculated due to the binary nucleation of $H_2SO_4$ and $H_2O$ vapor. The nucleation rate is a parameterized function of temperature, relative humidity, and the vapor-phase $H_2SO_4$ concentration, following the work of Kulmala et al. (1998), and the new particles are assumed to be 2.0 nm diameter. Recent studies have shown that organic vapors are involved in the nucleation process (Zhang et al., 2012) and further studies need to be conducted to consider the contributions of organic vapors to the nucleation process. The secondary organic aerosol (SOA) formation is simulated using a non-traditional SOA model including the volatility basis-set modeling method in which primary organic components are assumed to be semi-volatile and photochemically reactive and are distributed in logarithmically spaced volatility bins (Li et al., 2011a). The contributions of glyoxal and methylglyoxal to the SOA formation are also included in the SOA module. The SOA formation from glyoxal and methylglyoxal is parameterized as a first-order irreversible uptake by aerosol particles, with a reactive uptake coefficient of $3.7 \times 10^{-3}$ for glyoxal and methylglyoxal (Zhao et al., 2006). The simulation of inorganic aerosols in the WRF-CHEM model adopts the ISORROPIA Version 1.7 (Nenes et al., 1998).*"

We have also clarified in Supplementary Information (SI)-Section 1.1: "*The WRF-CHEM used in this study includes a new flexible gas phase chemical module and the CMAQ aerosol module developed by US EPA (Li et al., 2010; Binkowski and Roselle, 2003). In this aerosol component, the particle size distribution is represented as the superposition of three*

*lognormal sub-distributions, called modes. The processes of coagulation, particles growth by the addition of mass, and new particle formation are included."*

**3 Comment:** Also, aerosol impacts on meteorological fields could be significant, which might further affect the aerosol pollution condition in the lower troposphere. Also, aerosol-cloud interactions might modify temperature and moisture profiles and precipitation (Wang et al., Atmos. Chem. Phys. 11, 12421, 2011), leading to potential feedback on the atmospheric chemistry. Aerosol radiative effects induced by black carbon (BC) or other aerosol components could stabilize boundary layer and thus reduce the height of boundary layer, tending to exacerbate aerosol pollution near ground (Wang et al., Atmos. Environ. 81, 713, 2013). A particular important aspect is the aging of BC, which considerably enhances light absorption (Khalizov et al., J. Phys. Chem. 113, 1066, 2009; Peng et al., Proc. Natl. Acad. Sci. USA 113, 4266, 2016).

**Response:** We have clarified in Section 4: "*It is worth noting that interactions between the air pollution in China and ASM are two-way and their relationships are complicated and interrelated, especially with regard to the aerosol-meteorology interaction. Aerosol impacts on meteorology is significant due to its direct and indirect effects, which further influence the air pollution condition in the lower troposphere. Aerosol semi-direct effect induced by the light absorbing aerosols in the atmosphere stabilizes planetary boundary layer (PBL) and thus reduces the PBL height to exacerbate accumulation of air pollutants within the PBL, particularly for the aging process of black carbon which considerably enhances light absorption (Wang et al., 2013; Khalizov et al., 2009; Peng et al., 2016). In addition, aerosol plays an important role in the process of cloud formation and precipitation via acting as cloud condensation nuclei (CCN) and ice nuclei (IC). Therefore, aerosol-cloud interactions modify temperature and moisture profiles and influence precipitation, leading to potential feedback on the atmospheric chemistry (Wang et al., 2011).*".

**4 Comment:** It would also be necessary to mention the potential impacts of climate changes on pollution conditions in China (Wu et al., Sci. China: Earth Sci. 59, 1–16, 2016).

**Response:** We have clarified in Section 4: "*In addition, the ASM substantially influence spatial characteristics of the air pollutants transport and distribution in Eastern China on seasonal, inter-annual, and decadal scales (Wu et al., 2016). Further studies need to be*

*performed to investigate the impacts of the ASM variation on the air pollutants transport, which is modulated by climate changes.*".

**Minor points:**

**1 Comment**: As indicated in lines 81-84, the impacts of ASM on the air pollution over Northern China varies with the intensity of ASM. A case study on one-year monsoon season (May 2015) as reported in this work may not represent the various response under different ASM conditions. In addition to carry on more ASM episodes in future, how strong is the ASM season in this work relative to other years and/or the normal situation? A more detailed description of the strength of the simulated ASM will help us to evaluate the uncertainty range of the results in this study.

**Response:** We have clarified in Section 3.1: "*It is worth noting that the intensity of ASM substantially influences the temporal variation and spatial distribution of air pollutants (Wu et al., 2016). The East Asia summer monsoon index proposed by Zhang et al. (2003) is defined as a difference of anomalous zonal wind between the (10°-20°N, 100°-150°E) and (25°-35°N, 100°-150°E) at 850hPa during summer (June-August). The year of monsoon index greater than or equal to 2 is defined as the strong summer monsoon year, and the year of monsoon index less than or equal to -2 is defined as the weak summer monsoon year. The monsoon index calculated by China Meteorological Administration shows that the intensity of the summer monsoon in 2015 is close to the normals (SI-Figure S5).*".

**2 Comment**: It is good that the authors discuss the relative contribution of North China Plain to its surrounding regions in Figures 10 and 12, could the authors also provide the mean values of the contributions percentages in Tables 2 and 3? Also, please state the contribution percentages in the abstract. If available, could the authors add the uncertainty in the two tables and discuss it in the body text?

**Response:** We have revised Tables 2 and 3 in the manuscript.
We have clarified in Section 3.3.1: "*The impact of the NCP emissions on the daily average PM$_{2.5}$ concentration in the NEC and NWC from 22 to 28 May 2015 is summarized in Table 2. On average, the NCP emissions increase the PM$_{2.5}$ concentrations by 24.2, 9.6, 13.9, 6.5, and*

2.6 *μg m$^{-3}$ in Liaoning, Jilin, Shanxi, Shaanxi, and Inner Mongolia, with the average percentage contribution of 40.6%, 27.5%, 32.2%, 20.9%, and 16.7%, respectively.*"

We have clarified in Section 3.3.2: "*Table 3 summarizes the effects of the NCP emissions on the average afternoon O$_3$ concentration in the NEC and NWC from 22 to 28 May 2015.*", and "*On average, the NCP emissions distinctly increase the afternoon O$_3$ concentrations in Liaoning, Jilin, Shanxi, Shaanxi, and Inner Mongolia, with the average percentage contribution of 27.4%, 19.5%, 21.2%, 15.8%, and 8.0%, respectively (Table 3).*".

We have clarified the percentage contribution in the abstract: "*The average percentage contributions of the NCP emissions to the PM$_{2.5}$ level in Liaoning, Jilin, Shanxi, Shaanxi provinces are 40.6%, 27.5%, 32.2%, and 20.9%, respectively.*", and "*The average percentage contributions of the NCP emissions to the afternoon O$_3$ level in Liaoning, Jilin, Shanxi, and Shaanxi provinces are 27.4%, 19.5%, 21.2%, and 15.8%, respectively.*".

And we have also Clarified in Summary and Conclusions: "*Model results show that the NCP emissions contribute approximately an average of 24.2 and 13.9 μg m$^{-3}$ to the PM$_{2.5}$ concentration in Liaoning and Shanxi during the episode, with the average percentage contribution of 40.6% and 32.2%, respectively. The NCP emissions enhance the PM$_{2.5}$ level by 9.6 and 6.5 μg m$^{-3}$ in Jilin and Shaanxi on average, with the percentage contribution of 27.5% and 20.9%, respectively. The NCP emissions also substantially influence the O$_3$ concentration in the NEC and NWC. The NCP emissions increase the afternoon (12:00 - 18:00 BJT) O$_3$ concentration in Liaoning by 46.5 μg m$^{-3}$ on average during the episode, followed by 35.1 μg m$^{-3}$ in Shanxi, 28.7 μg m$^{-3}$ in Jilin, and 20.7 μg m$^{-3}$ in Shaanxi, with the average percentage contribution of 27.4%, 21.2%, 19.5%, and 15.8%, respectively.*".

Additionally, we have included the uncertainties (standard deviation) in Table 2 and Table 3 and also clarified in Section 3.3.2: "*Furthermore, it is worth noting that uncertainties from meteorological field simulations, emission inventories, and the chemical mechanism used in simulations, have large potentials to influence evaluation of the effect of the NCP emissions on the PM$_{2.5}$ and O$_3$ concentrations in the NEC and NWC (Carter and Atkinson, 1996; Lei et al., 2004; Song et al., 2009; Bei et al., 2017).*":

**3 Comment**: In lines 221-222, the work by Wang et al. (PNAS, 2016) is relevant, which has documented the possible efficient SO$_2$ conversion pathway with assistant of NO$_2$ in aqueous phase.

**Response:** We have clarified in Section 3.2.2: "*Wang et al. (2016) have also reported that the aqueous oxidation of $SO_2$ by $NO_2$ is important to the efficient sulfate formation.*".

**4 Comment:** Regarding the uncertainties from meteorological fields as mentioned in line 335, how do the simulations perform in predicting the regular meteorological parameters, such as temperature, wind speed, and so on, comparing to observations?

**Response:** We have clarified in SI-Section-4.1: "*Considering that the meteorological conditions play an important role in the dispersion or accumulation of air pollutants, simulated meteorological fields are compared to the observations. Figure S6 shows the temporal profiles of the simulated and observed surface temperature at the observation sites in Beijing, Tianjin, Shijiazhuang, Shanghai, and Hefei from 22 to 28 May 2015 (Figure S1). The WRF-CHEM model generally reproduces the temporal variation of the temperature during the study episode compared with the observations, with IOAs exceeding 0.65, but slightly underestimates the temperature in Shanghai and Hefei, particularly during the noontime, with MBs of -1.4 and -1.8 °C, respectively. The overestimation of the temperature exists in Beijing, Tianjin, and Shijiazhuang, with MBs of 3.3, 1.9, and 3.1 °C, respectively. Figure S7 presents the temporal profiles of the simulated and observed surface wind speed in the observation sites in Beijing, Tianjin, Shijiazhuang, Shanghai, and Hefei from 22 to 28 May 2015. In general, the model performs reasonably in predicting the temporal variation of the wind speed in these cities, particularly in Tianjin, with IOA of 0.64, but the simulated surface wind speed is still biased considerably due to the implication of building distributions and heights and the inability of the model for microscale simulations (Chen et al., 2011; Lee et al., 2011).*"

**5 Comment:** In section 3.3 lines 259-260, the authors mentioned that the simulations can be used for evaluating the interactions of the two emissions (i.e., with NCP emissions only and with non-NCP emissions only), but there are no discussions about the interactions in the remaining part of the manuscript. It is interesting to know how possible the non-NCP emissions affect NCP. Could the authors show some results about the interactions of the emissions from the two regions?

**Response:** We have clarified in SI-Section-5: "*Table S4 summarizes the contribution of*

*interactions between NCP and non-NCP emissions to the daily average PM$_{2.5}$ concentration in the NCP, NEC and NWC from 22 to 28 May 2015. The interaction between NCP and non-NCP emissions generally increases the PM$_{2.5}$ concentration due to the enhancement of precursors of air pollutants and the aerosol radiation feedback. The average contribution of interactions between NCP and non-NCP emissions to the PM$_{2.5}$ concentration in the NCP, Jilin, Liaoning, Shanxi, Shaanxi and Inner Mongolia is 3.0, 2.6, 7.9, 1.9, 1.7, and 0.8 µg m$^{-3}$, or 4.6%, 7.5%, 13.3%, 4.4%, 5.5%, and 5.1%, respectively. The contribution of interactions between NCP and non-NCP emissions on the daily afternoon average O$_3$ concentration in the NCP, NEC and NWC from 22 to 28 May 2015 is summarized in Table S5. On average, the interaction of these two emissions increases the afternoon O$_3$ concentrations by 16.9, 12.8, 17.9, 12.6, 11.1, and 5.8 µg m$^{-3}$, or 10.5%, 8.7%, 8.2%, 7.6%, 8.5%, and 5.5%, in the NCP, Jilin, Liaoning, Shanxi, Shaanxi and Inner Mongolia, respectively.".*